# A luciferase prosubstrate and a red bioluminescent calcium indicator for imaging neuronal activity in mice

Xiaodong Tian[1,2], Yiyu Zhang[1,2], Xinyu Li[1,2,4], Ying Xiong[1,2], Tianchen Wu[1,2] & Hui-Wang Ai [1,2,3✉]

Although fluorescent indicators have been broadly utilized for monitoring bioactivities, fluorescence imaging, when applied to mammals, is limited to superficial targets or requires invasive surgical procedures. Thus, there is emerging interest in developing bioluminescent indicators for noninvasive mammalian imaging. Bioluminescence imaging (BLI) of neuronal activity is highly desired but hindered by insufficient photons needed to digitalize fast brain activities. In this work, we develop a luciferase prosubstrate deliverable at an increased dose and activated in vivo by nonspecific esterase. We further engineer a bright, bioluminescent indicator with robust responsiveness to calcium ions ($Ca^{2+}$) and appreciable emission above 600 nm. Integration of these advantageous components enables the imaging of the activity of neuronal ensembles in awake mice minimally invasively with excellent signal-to-background and subsecond temporal resolution. This study thus establishes a paradigm for studying brain function in health and disease.

[1] Department of Molecular Physiology and Biological Physics, University of Virginia, Charlottesville, VA 22908, USA. [2] Center for Membrane and Cell Physiology, University of Virginia, Charlottesville, VA 22908, USA. [3] The UVA Comprehensive Cancer Center, University of Virginia, Charlottesville, VA 22908, USA. [4]Present address: State Key Laboratory of Reproductive Regulation and Breeding of Grassland Livestock, School of Life Sciences, Inner Mongolia University, Hohhot, China. ✉email: huiwang.ai@virginia.edu

Fluorescence imaging is the standard method for following brain activity in small behaving animals. Genetically encoded fluorescent indicators, including GCaMPs and other $Ca^{2+}$, voltage, and neurotransmitter indicators, allowed the tracking of neuronal activities in specific brain regions and cell types in mammals with the high spatiotemporal resolution for extended periods[1–3]. Despite the progress, fluorescence neuronal imaging is practically invasive and only reaches a shallow depth. Cranial windows or thinned skulls are often used to access the cortex. Due to tissue absorption and scattering, the imaging depth is ~200 μm for widefield one-photon and a couple of millimeters with multiphoton excitation[3–5]. To reach deeper brain regions, more invasive procedures, such as implanting optical fibers or gradient-index (GRIN) lenses, are needed[3].

Bioluminescence, which refers to photon emission from luciferase-catalyzed exothermic oxidation of the corresponding luciferin, is a promising imaging modality for noninvasive in vivo recording[6,7]. Because bioluminescence needs no excitation, photons emitting from the embedded light sources can travel through several centimeters of mammalian tissue[8]. In addition, compared to fluorescence, BLI has a low background, no photobleaching and phototoxicity, and minimized disturbances of light-sensitive biological components (e.g., the circadian system)[9–12]. Moreover, BLI has excellent compatibility with popular optogenetic and optochemical tools[13]. Macroscopic BLI only requires a dark box with a sensitive camera. Similar setups are already in many laboratories and core facilities; otherwise, they are commercially available or can be constructed at affordable costs.

Commonly used luciferase-luciferin pairs originate from either insects or marine organisms[6,7]. Insect luciferases are generally slow in catalysis and consume adenosine triphosphate (ATP) for luciferin activation and photon production. In contrast, the oxidation of the coelenterazine (CTZ; Supplementary Fig. 1) luciferin by marine luciferases is ATP-independent. NanoLuc, a marine luciferase variant, exhibits a high photon production rate in the presence of furimazine (FRZ; Supplementary Fig. 1), a synthetic CTZ analog[14]. However, NanoLuc has several unfavorable features for in vivo BLI, including low tissue penetration of its blue emission, and limited substrate solubility and stability. Recent studies have partially addressed these issues by developing additional CTZ analogs and NanoLuc mutants[15–18] or genetically fusing NanoLuc to long-wavelength-emitting fluorescent proteins (FPs) for redder emission via bioluminescence resonance energy transfer (BRET)[15,16,19,20].

Bioluminescent indicators that change signals in response to neuronal activity are needed for functional imaging. $Ca^{2+}$ is a ubiquitous second messenger, and intracellular $Ca^{2+}$ has been used as a proxy for neuronal activity[1–3]. Previous studies have used $Ca^{2+}$-sensitive photoproteins, such as aequorin, obelin, and their mutants, for $Ca^{2+}$ detection, but they have drawbacks, including weak light emission and irreversible responses due to slow luciferin recharging[21–23]. Other studies introduced $Ca^{2+}$-sensory elements into luciferases, including NanoLuc and NanoLuc-FP hybrid reporters, resulting in reversible $Ca^{2+}$ indicators with much-increased light production[13,24–27].

Despite the progress, functional BLI (fBLI) of neuronal activity, which requires fast digitalization, is still hindered by insufficient photons reaching detectors. First, the intrinsic photon production rates of NanoLuc and NanoLuc-based indicators remain several orders of magnitude lower than those achievable in typical fluorescence imaging setups[28,29]. Another limiting factor is the low amount of marine luciferase substrates that can be systematically delivered to the brain[17,30,31]. Moreover, most current bioluminescent $Ca^{2+}$ indicators (Table 1) emit short-wavelength light strongly attenuated by brain tissue, skull, and skin[13,25–27]. Orange CaMBIs, which were created by inserting the $Ca^{2+}$-sensory calmodulin (CaM) and M13 moieties between residues 133 and 134 of NanoLuc linked to two copies of CyOFP1 (an orange-emitting FP), are the only NanoLuc-based indicators with appreciable emission above 600 nm[29]. The Orange CaMBI 110 (OCaMBI110) variant has been successfully used to image in vivo $Ca^{2+}$ dynamics in the mouse liver but not yet in the brain[17,29].

To fill the fBLI technical gap, we develop a luciferase prosubstrate for enhanced luciferin delivery and a bioluminescent indicator with redder emission and remarkable brightness and $Ca^{2+}$ responsiveness. We characterize this integrated system for imaging $Ca^{2+}$ dynamics in cultured cell lines, primary mouse neurons, and acute brain slices. Moreover, because of the drastically increased photon flux, the integrated system enables the minimally invasive imaging of deep-brain $Ca^{2+}$ dynamics in awake mice in response to behavioral and disease triggers.

## Results

**Design and chemical synthesis of luciferase prosubstrates.** Our group previously reported teLuc, a bright and red-shifted NanoLuc mutant, and its paired DTZ substrate (Supplementary Fig. 1), which showed promising BLI performance in mice[15]. DTZ could be synthesized from inexpensive commercial reagents in two steps with excellent yields (Supplementary Fig. 2). We thus aimed to chemically modify DTZ to enhance its delivery to the brain. First, we compared the computed logD (the octanol/water

**Table 1 Properties of bioluminescence $Ca^{2+}$ indicators based on NanoLuc or NanoLuc-derived luciferases[a].**

| Construct | Peak emission (nm)[b] | BL/$BL_0$[c] | $K_d$ (nM) | Emission fraction > 600 nm | BRET Donor | BRET Acceptor | $Ca^{2+}$ sensing domain | Reference |
|---|---|---|---|---|---|---|---|---|
| GeNLs($Ca^{2+}$) | 517 | 3.9–5 | 60–520 | 0.017[d] | NLuc split at residue 66 | mNeonGreen | CaM, M13 | 25 |
| CalFluxVTN | 525 | 5[e] | 480 | 0.04[d] | NLuc | Venus | TnC | 13 |
| LUCI-GECO1 | 515 | 2.6[e] | 285 | <0.02 | NLuc | ncpGCaMP6s | CaM, RS20 | 26 |
| GLICO | 515 | 23 | 230 | <0.02 | NanoBiT | GCaMP6f | CaM, RS20 | 27 |
| ReLICO | 452 | 3.4 | $1.5 \times 10^6$ | <0.02 | NanoBiT | R-CEPIA1er | CaM, RS20 | 27 |
| Orange CaMBIs | 586 | 7–8[f] | 110–300 | 0.33[d] | NLuc split at residue 133 | CyOFP1 | CaM, M13 | 29 |
| BRIC | 595 | 6.5 | 133 | 0.54 | teLuc split at residue 133 | mScarlet-I | CaM, M13 | This work |

[a]Data for BRIC were determined in this work. Unless otherwise indicated, other data were reported or calculated from graphs in the initial publications. [b]Wavelength for the major or the most red-shifted emission peak. [c]Intensity ratio with or without $Ca^{2+}$ at the indicated peak emission wavelength. [d]Adapted from Reference [29]. [e]Originally described as green/blue ratiometric indicators, and the response range (R/$R_0$) was reported to be 11 and 5 for CalFluxVTN and LUCI-GECO, respectively. [f]Our determined value is ~2.5–3.7 for OCaMBI110. The discrepancy may be caused by expression and assay conditions and the variable oligomerization states of Orange CAMBIs.

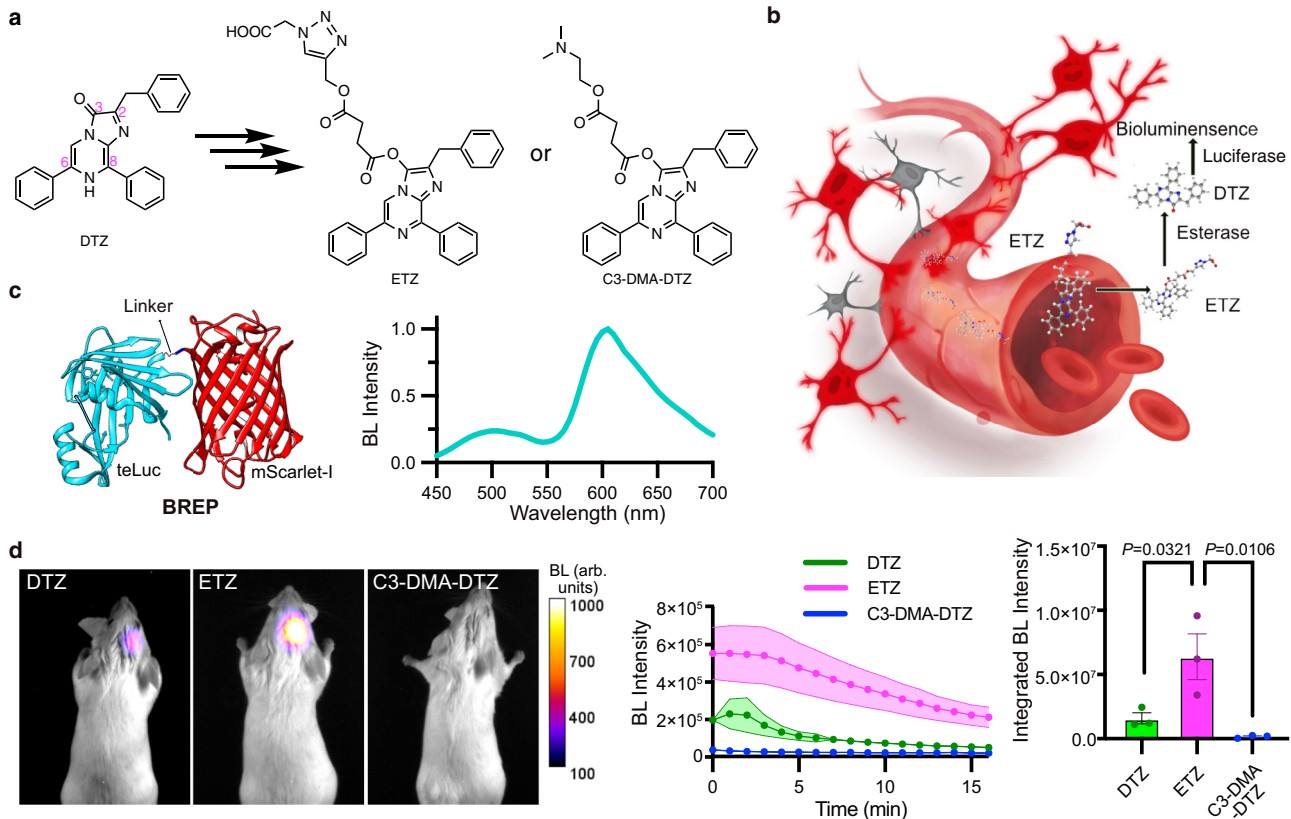

**Fig. 1 Luciferase prosubstrates and initial evaluation for brain imaging with BREP. a** Illustration of the derivatization of DTZ to generate two luciferase prosubstrates (ETZ and C3-DMA-DTZ) with modulated solubility, signal duration, and in vivo delivery capability. **b** Proposed conversion of ETZ to DTZ in vivo by nonspecific esterase, followed by luciferase-catalyzed oxidation of DTZ to generate bioluminescence. **c** Schematic illustration of the domain arrangement of BREP (left) and its bioluminescence emission spectrum in the presence of DTZ (right). **d** Left: Representative bioluminescence images of live mice with BREP-expressing HEK 293 T cells stereotactically injected into the hippocampus. The substrates were administered via tail vein at their respective saturation concentrations. Images with peak bioluminescence intensities were presented in pseudocolor overlaid on corresponding brightfield images. Middle: Bioluminescence intensity over time shown for each substrate. Right: Comparison of the integrated bioluminescence intensity (area under the curve) with the residual background subtracted. Data are presented as mean ± s.e.m. ($n = 3$ mice). $P$ values were derived from ordinary one-way ANOVA followed by Dunnett's multiple comparisons test. BL, bioluminescence. Arb. units, arbitrary units. Source data are provided as a Source Data file.

distribution coefficient) of DTZ (4.5 at pH 7) with common blood-brain barrier (BBB)-permeable drugs[32]. Decreasing the lipophilicity of DTZ was suggested to increase its delivery to the brain. Furthermore, although the mechanisms limiting the peripheral delivery of CTZ and its analogs to the brain are not fully understood, the BBB efflux transporters, such as P-glycoprotein (P-gp) and ABCG2, were shown to pump out CTZ[33,34]. The BBB efflux issue is further compounded by additional unfavorable factors, such as the rapid clearance and low solubility of these substrates[30,35]. The literature reported that adding a succinate group to paclitaxel (Taxol) reduced the interaction with P-gp and increased brain distribution vastly[36,37], because P-gp unfavorably interacts with negatively charged molecules and the succinate addition installs a carboxylate functional group with a $pK_a$ of ~4. Thus, we chose to use the carboxylate functional group to modify DTZ with the hope of enhancing its hydrophilicity, reducing BBB efflux, and increasing possible dosage via aqueous intravenous injection buffers.

Because the C3 carbonyl group of the DTZ imidazopyrazine ring (Fig. 1a) is required for substrate oxidation[6], C3 derivatizations will generate caged substrates resistant to auto- and luciferase-catalyzed oxidation. We designed and synthesized a compound with an extended carboxylate via C3 (see ETZ in Fig. 1a, Supplementary Figs. 1 and 2; computed logD is 1.1 at pH 7). When ETZ is delivered in vivo, nonspecific esterase is expected to hydrolyze the ester bonds,

resulting in free DTZ for luciferase-catalyzed bioluminescence (Fig. 1b). We named this compound ETZ for esterase-dependent activation and enhanced in vivo performance (presented below). In addition, for comparison purposes, we synthesized C3-DMA-DTZ (Fig. 1a, Supplementary Figs. 1 and 2; computed logD is 3.8 at pH 7), which contains ester linkages but is partially positively charged at physiological pH.

**Engineering of a BRET-based bioluminescent red protein.** Photons within the optical window (600–950 nm) are less absorbed and scattered by mammalian tissue, thus penetrating tissue deeper[38]. It is an established strategy to shift the bioluminescence of marine luciferases toward the optical window via BRET to FPs[15,16,19,20]. Our previous study fused a LumiLuc luciferase to a bright red FP (RFP) mScarlet-I, resulting in a LumiScarlet reporter with ~ 51% of the total emission above 600 nm[16]. Following the success, we created a similar fusion between teLuc and mScarlet-I and optimized the linker for increased BRET efficiency (Supplementary Fig. 3). As a result, we arrived at a bright Bioluminescent Red Protein (BREP) with ~60% of its total emission above 600 nm (Fig. 1c). Although the emission of teLuc is less red-shifted than LumiLuc and has less spectral overlap with mScarlet-I, we observed more efficient BRET in BREP than LumiScarlet due to the shorter

donor-acceptor distance and a possible spatial orientation favoring donor-acceptor dipole coupling in BREP[39].

**Comparison of luciferase prosubstrates in mammalian cells and for brain imaging in mice.** We next compared DTZ, ETZ, and C3-DMA-DTZ for bioluminescence in cultured mammalian cells. We transiently expressed BREP in human embryonic kidney (HEK) 293 T cells and imaged the cells in 96-well plates upon adding 25 μM of each compound (Supplementary Fig. 4). The initial bioluminescence of ETZ-treated cells was lower than DTZ-treated cells but slightly higher than cells treated with C3-DMA-DTZ. Because the signals of the DTZ and C3-DMA-DTZ groups decayed quickly while ETZ allowed sustained bioluminescence, the integrated signal of the ETZ group during the examined 50 min period was ~2- and 8-fold of the DTZ and C3-DMA-DTZ groups, respectively. The results suggest that ETZ is a promising substrate for mammalian imaging applications.

We further compared the three compounds for brain delivery in mice. We injected ~7,000 HEK 293 T cells transiently transfected with BREP to the hippocampus in anesthetized BALB/cJ mice. After intracranial cell injection, we immediately administered 100 μL buffers containing each substrate at their saturation concentrations via tail vein. ETZ exhibited better solubility than DTZ and C3-DMA-DTZ (Supplementary Fig. 5a) and could be delivered at a dosage of 0.68 μmol per mouse. Mice infused with ETZ showed the most robust and durable bioluminescence (Fig. 1d and Supplementary Fig. 5b). During the 16-min window used in this experiment, the integrated signal of the ETZ group was ~4-fold and ~39-fold higher than the DTZ and C3-DMA-DTZ groups, respectively.

Because there were concerns that endogenous neurons may be different from implanted HEK 293 T cells in compound uptake and esterase activity, we further prepared adeno-associated viruses (AAVs) with the BREP gene driven by the human synapsin I (hSyn) promoter and transduced hippocampal neurons in live mice via stereotactic injection. We tested the BREP-expressing mice against a panel of CTZ analogs (Supplementary Fig. 6). FFz (Supplementary Fig. 1) was recently reported as a NanoLuc substrate with enhanced in vivo performance[17], so we chemically synthesized FFz for comparison. When the compounds were delivered at their saturation concentrations, ETZ again surpassed all others by generating the brightest and most durable emission in mice. In terms of bioluminescence integrated over the experimental period, mice infused with ETZ were ~4.2-, 8.8-, 67.3-, and 122.7-fold of DTZ, C3-DMA-DTZ, FFz, and FRZ, respectively. Therefore, ETZ was confirmed as a superior luciferase substrate for brain imaging in mice.

We further compared the bioluminescence of the BREP-expressing mice infused with the equimolar amount (0.25 μmol) of ETZ and DTZ. Although mice in the ETZ group seemed slightly brighter than those in the DTZ group, there was inadequate statistical power to resolve the difference (Supplementary Fig. 6). Nevertheless, the result unambiguously supports that the enhanced in vivo performance of ETZ from DTZ was primarily caused by the increased substrate delivery enabled by the increased solubility.

In addition, we evaluated the background bioluminescence of ETZ in blank mice or mice with untransfected HEK 293 T cells stereotactically injected into the hippocampus. Corroborating our previous study on DTZ[40], the background signal from ETZ was negligible compared to authentic bioluminescence signals (Supplementary Fig. 7).

**Engineering and initial characterization of a bioluminescent red indicator for Ca$^{2+}$.** We further engineered a bioluminescent Ca$^{2+}$ indicator from BREP. Inspired by Orange CaMBIs, we similarly inserted CaM and M13 between residues 133 and 134 of teLuc in BREP, resulting in a prototype showing a 2.5-fold (BL/BL$_0$) Ca$^{2+}$-dependent bioluminescence increase (Supplementary Fig. 8). Next, we performed six rounds of error-prone PCRs and screened the libraries for high bioluminescence brightness and Ca$^{2+}$ responsiveness. The effort led to a Bioluminescent Red Indicator for Ca$^{2+}$ (BRIC) with a 6.5-fold (BL/BL$_0$) response (Fig. 2a, b, Supplementary Fig. 9, and Table 1). BRIC, at its Ca$^{2+}$-bound condition, retained ~ ~ 46% of the brightness of BREP. Using the purified protein, the dissociation constant ($K_d$) of BRIC to Ca$^{2+}$ was determined to be 133 nM (Fig. 2c). The response magnitude increased as pH changed from 5.5 to 7 and was relatively stable at pH > 7 (Fig. 2d). When compared in parallel, BRIC was more responsive to Ca$^{2+}$ than OCaMBI110 (Supplementary Fig. 10a, b and Table 1).

To evaluate BRIC for imaging cellular Ca$^{2+}$ dynamics, we transiently expressed BRIC in human cervical cancer HeLa cells, in which histamine can evoke Ca$^{2+}$ waves[41]. As expected, we observed single-cell bioluminescence oscillations in response to histamine under bioluminescence microscopy (Fig. 2e and Supplementary Movie 1). Furthermore, we transduced cultured primary mouse neurons with BRIC AAVs harboring the hSyn promoter and successfully detected Ca$^{2+}$ influx after high K$^+$ depolarization (Fig. 2f and Supplementary Movie 2). In contrast, the control BREP-expressing neurons showed little bioluminescence increase.

In addition, we compared BRIC with OCaMBI110 in HeLa cells and cultured mouse neurons. Under fluorescence channels, we observed extensive puncta in cells overexpressing OCaMBI110 (Supplementary Fig. 10c) but not in BRIC-expressing cells. In addition, the cells with fluorescent puncta showed little bioluminescence and were unresponsive to histamine or high K$^+$. Although the exact reason is unknown, the observed puncta may be OCaMBI110 oligomers because each OCaMBI110 molecule contains two copies of dimeric CyOFP1 (Supplementary Fig. 10d)[19]. Moreover, with punctum-containing cells excluded from analysis, the response magnitude of BRIC was still higher than OCaMBI110 in both HeLa cells and cultured neurons (Supplementary Fig. 10e, f).

Further, we examined BRIC for imaging Ca$^{2+}$ influx in primary mouse neurons induced with ionomycin, a Ca$^{2+}$-selective ionophore (Supplementary Fig. 11)[42]. The bioluminescence of cultured neurons in Ca$^{2+}$-containing buffers increased upon the addition of ionomycin. The responses depended on extracellular Ca$^{2+}$ concentrations, as higher Ca$^{2+}$ caused faster and more pronounced intensity changes.

**Imaging of the bioluminescent Ca$^{2+}$ indicator in the brain in live mice.** Because ETZ exhibited enhanced bioluminescence in the brain in live mice and BRIC performed well in cultured cells, we next examined the integration of ETZ and BRIC for in vivo BLI. We started by investigating the brightness of BRIC and ETZ in the brain in live mice. We stereotactically injected BRIC and OCaMBI110 AAVs (adjusted to the same viral titer) into the hippocampus of BALB/cJ mice (Fig. 3a) and compared the brightness of the two indicators on day 19 post viral administration. We examined OCaMBI110 with either FRZ or the recently reported FFz substrate. Each substrate was intravenously administered to anesthetized animals at their saturation concentrations (Supplementary Fig. 12a). Using an EMCCD camera, we followed the signals of BRIC with 500-ms exposure for 1 h. The starting bioluminescence of BRIC in the presence of ETZ was ~168- and 29-fold higher than OCaMBI110 in the presence of FRZ and FFz, respectively (Fig. 3b and Supplementary Fig. 12b). In addition, the BRIC signals were consistently higher than the background during periods much longer than

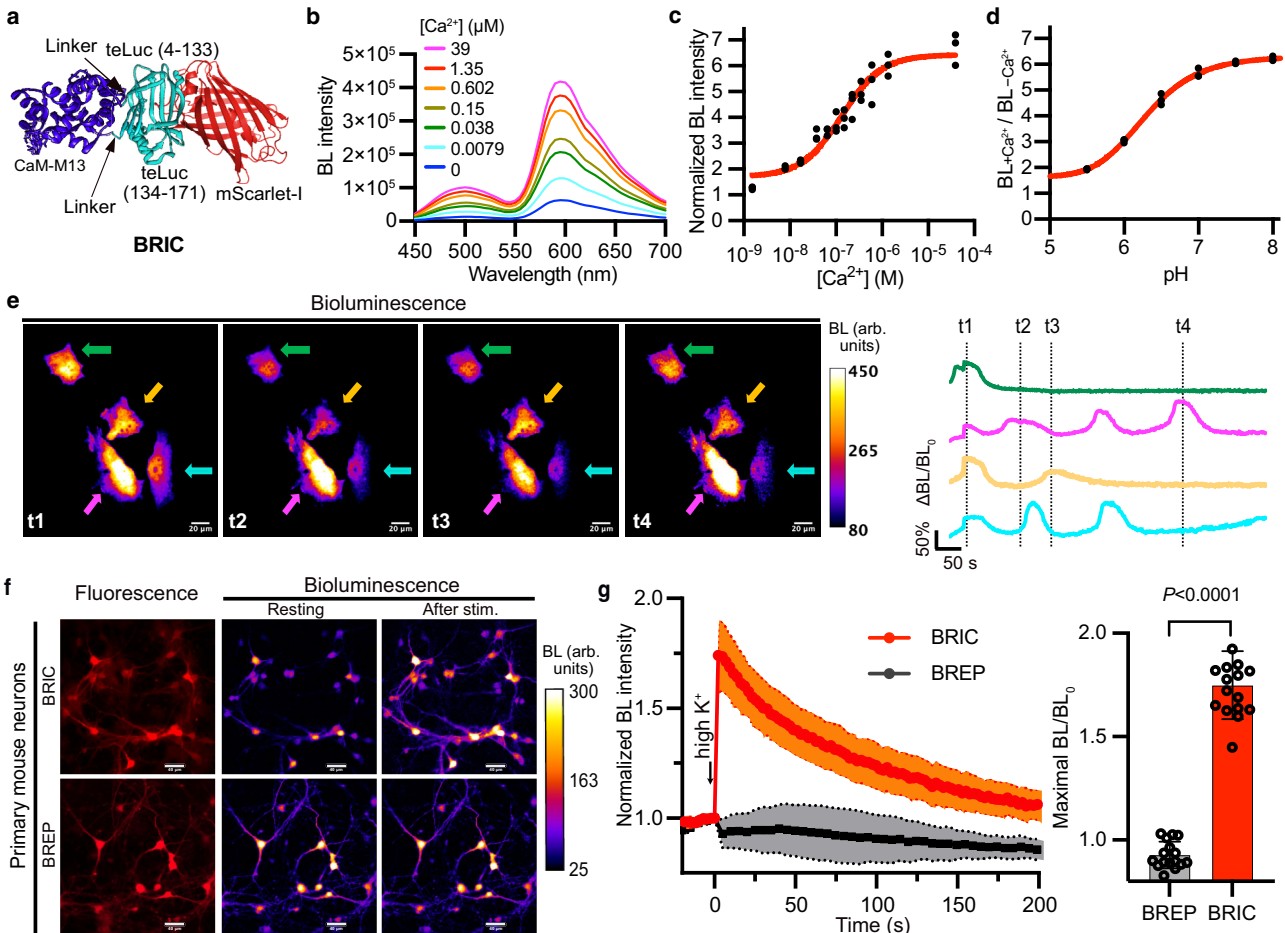

**Fig. 2 Characterization of BRIC in vitro and cultured cells. a** Schematic illustration of the domain arrangement of BRIC. **b** Bioluminescence spectra of BRIC in the presence of DTZ and indicated concentrations of free $Ca^{2+}$. **c** $Ca^{2+}$-dependency of BRIC bioluminescence at the peak emission wavelength (595 nm). $n = 3$ technical repeats. A one-site binding model was used to fit the data and derive the dissociation constant ($K_d = 133 \pm 24$ nM). **d** Ratios of BRIC bioluminescence in the presence (39 μM) to the absence of $Ca^{2+}$ across the indicated pH range. $n = 3$ technical repeats. **e** Representative pseudocolored bioluminescence images (left) and intensity traces (right) of histamine-induced $Ca^{2+}$ dynamics in HeLa cells. Arrows indicate individual cells, and the colors of the arrows are identical to the colors of the intensity traces. The baselines of the intensity traces were corrected for monoexponential decay caused by substrate consumption. The experiment was repeated five times with similar results. Scale bar, 20 μm. **f** Representative fluorescence and bioluminescence images of BRIC- or BREP-expressing primary mouse neurons. High $K^+$ (30 mM) was used to depolarize the neurons. Scale bar, 40 μm. **g** Quantification of bioluminescence intensity changes of neurons in response to high $K^+$. The baselines were corrected using a monoexponential decay model. Data are presented as mean ± s.d. ($n = 16$ cells), and the $P$ value was derived from unpaired two-tailed $t$-tests. The GraphPad Prism software does not provide extract $P$ values below 0.0001. BL, bioluminescence. Arb. units, arbitrary units. Source data are provided as a Source Data file.

OCaMBI110 with either substrate (Fig. 3c and Supplementary Fig. 12c). Regarding the signals integrated over time, BRIC was ~ 153.7- and 22.0-fold of OCaMBI110 in the presence of FRZ and FFz, respectively (Fig. 3d). Furthermore, we prepared acute brain slices from BRIC-expressing mice and observed bioluminescence rise in response to high $K^+$, confirming the activity of BRIC in the brain tissue (Supplementary Fig. 13 and Supplementary Movie 3).

We finally examined BRIC for monitoring $Ca^{2+}$ dynamics in awake mice (Fig. 4a). First, we administered the virus to the basolateral amygdala (BLA) region (Supplementary Fig. 14) responsible for fear processing[43]. Upon the intravenous injection of ETZ, head-fixed mice were subjected to BLI in a dark box. Meanwhile, we applied electric footshock repeatedly to mice. In response to the footshock stimuli, we observed reproducible bioluminescence increase in BRIC-expressing mice (Fig. 4b and Supplementary Movie 4). BREP was included as a negative control. BREP-expressing mice showed very small responses to

footshock, probably caused by animal motions or blood flow changes. In addition, the indicator was expressed in the hippocampus of C57BL/6 J mice, and kainic acid (KA) was used to induce epileptic seizures, a condition known for abnormal hippocampal $Ca^{2+}$ waves[44,45]. We monitored the tremor behavior of KA-treated mice, which were next infused with ETZ via tail vein before the head-fixed animals were imaged in the dark box. As expected, we detected rapid bioluminescence intensity changes in BRIC-expressing mice, which were statistically different from BREP-expressing control mice (Fig. 4c and Supplementary Movie 5). However, there were noticeable response variations between individual BRIC-expressing animals. We reason that the differences may reflect biological variations since the extent of KA-induced seizures and calcium responses could be different in individual animals, as well as technical limitations since KA-induced seizures are known to be intermittent and some mice were probably not imaged right during the occurrence of seizures[44,45].

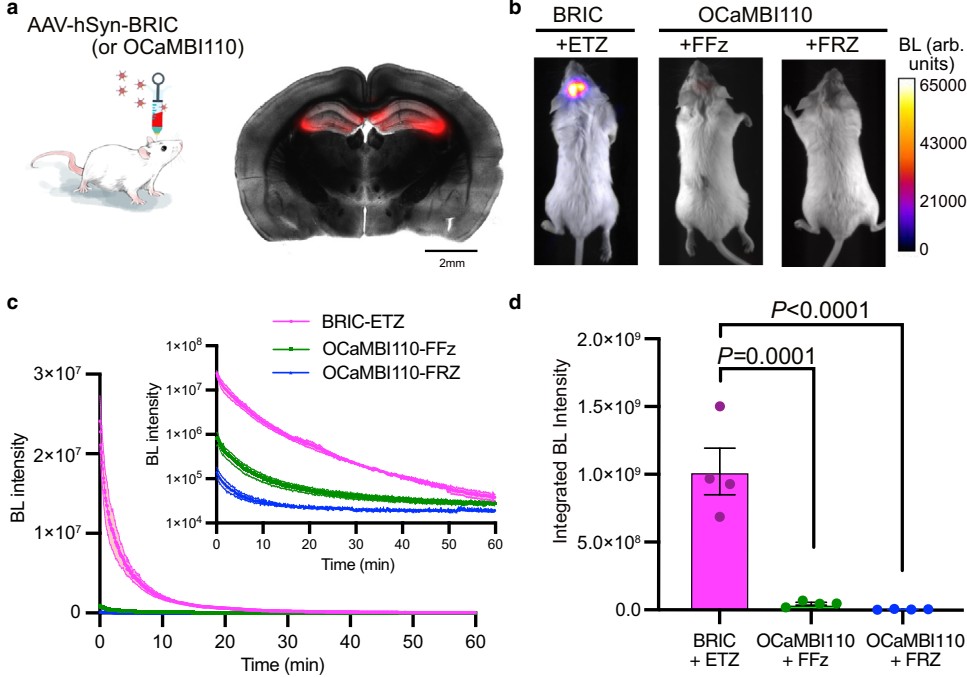

**Fig. 3 Brightness comparison of BRIC and Orange CaMBI 110 (OCaMBI110) in the hippocampus in live mice. a** Left: Illustration of stereotactic intracranial administration of AAVs. Right: Image of an acute brain slice prepared from a BRIC-transduced mouse, showing the successful expression of the indicator in the hippocampus. The fluorescence channel (red) is overlaid on the corresponding grayscale brightfield image. Scale bar, 2 mm. **b** Representative bioluminescence images of live mice with the hippocampus transduced with BRIC or OCaMBI110 AAVs. The substrates were administered via tail vein at their respective saturation concentrations. Images with peak bioluminescence intensities were presented in pseudocolor overlaid on corresponding brightfield images. **c** Bioluminescence intensity over time presented for each substrate. The inset shows the same results with a logarithmic y-axis. **d** Comparison of the integrated bioluminescence intensity (area under the curve) with the residual background subtracted. Data are presented as mean ± s.e.m. ($n = 4$ mice). $P$ values were derived from ordinary one-way ANOVA followed by Dunnett's multiple comparisons test. The GraphPad Prism software does not provide extract $P$ values below 0.0001. BL, bioluminescence. Arb. units, arbitrary units. Source data are provided as a Source Data file.

## Discussion

In summary, we have developed an integrated fBLI platform for imaging brain $Ca^{2+}$ dynamics in awake mice. We first enhanced the solubility and in vivo delivery of the luciferin via a prosubstrate strategy. We next developed BREP, a luciferase-FP fusion reporter with remarkable emission above 600 nm. From BREP, we further engineered a bioluminescent $Ca^{2+}$ indicator with high brightness and $Ca^{2+}$ responsiveness. We validated the bioluminescent $Ca^{2+}$ indicator for imaging $Ca^{2+}$ dynamics in cultured cells and brain slices. Finally, we combined the prosubstrate and the $Ca^{2+}$ indicator for imaging the activity of neuronal populations in live animals.

Previous studies used CTZ to activate luminopsins (luciferase-channelrhodopsin fusions) in the brain in live mice[46], but a fast clearance of CTZ from the brain was advantageous in such cases since the quick clearance facilitated temporal control. In this study, we introduced a carboxylate via the C3 position of the DTZ imidazopyrazine ring, leading to the ETZ prosubstrate exhibiting resistance to auto-oxidation, increased aqueous solubility, and slowed live-cell signal decay. When administered to live cells or mice, ETZ is expected to be hydrolyzed by esterase to liberate DTZ[47,48]. Since significant carboxylesterase activities were detected in mouse liver and blood[47,49], the labile ester linkages of ETZ may break down before it crosses the BBB. Indeed, our imaging results of BREP-expressing mice infused with an equimolar amount of ETZ and DTZ suggest that the increased solubility and dosage were the main factors for the enhanced in vivo performance of ETZ, even though the BBB efflux issue inspired our initial substrate design. Future studies may examine

luciferins with a stable carboxylate functional group, but it will require further engineering of corresponding luciferases and indicators to accommodate the new substrates.

Of note, several other studies developed C-3 caged CTZ or FRZ analogs, some of which also slowed bioluminescence decays in living cells[50,51]. Particularly, a substrate manufactured by Promega Corporation, namely ViviRen, was shown to improve the light output of *Renilla* luciferase in live mice bearing brain tumors compared to the equimolar amount of CTZ[30]. However, these previously explored C-3 caging groups were typically hydrophobic, unfavorably reducing substrate solubility in aqueous solutions. Also, a prosubstrate strategy was recently used to deliver the amide derivatives of firefly luciferins into the mouse brain for fatty acid amide hydrolase (FAAH)-dependent activation[52,53].

BREP is the most red-shifted marine luciferase-FP fusion protein that retains high brightness. The construct uses mScarlet-I as the BRET acceptor. Because existing far-red and near-infrared FPs that are more red-shifted than mScarlet-I have much-reduced fluorescence quantum yields, it is not yet possible to use the FP fusion strategy to further red-shift bioluminescence without remarkably harming brightness. Alternatively, one may covalently conjugate far-red or near-infrared synthetic dyes to luciferase variants via self-labeling protein tags (e.g., Halo tag)[18,54,55].

BRIC is a high-performance and red-shifted bioluminescent $Ca^{2+}$ indicator. We compared it with the current benchmark, OCaMBI110. BRIC showed larger responses to $Ca^{2+}$ as purified proteins and in HeLa cells and primary neurons. In addition,

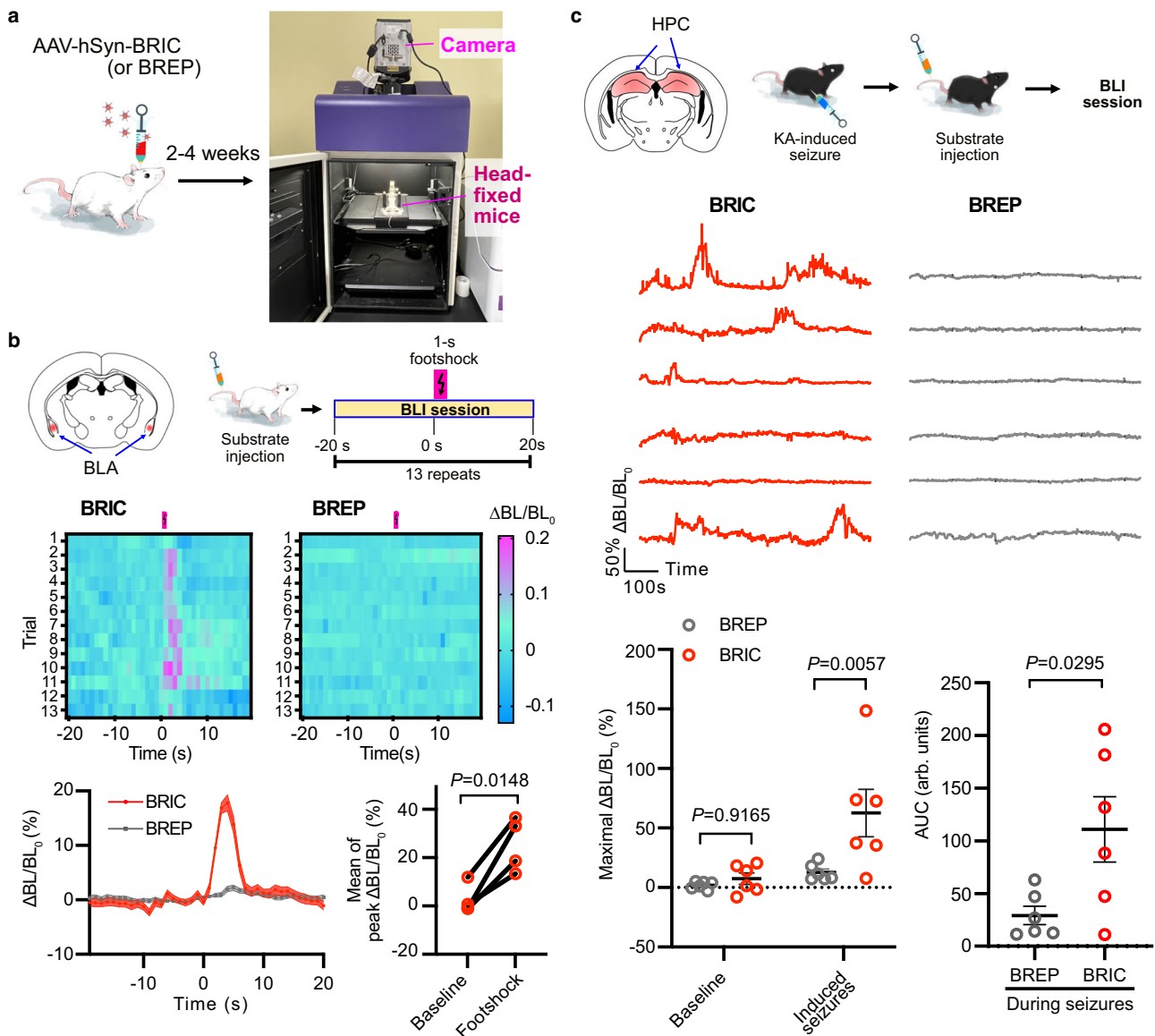

**Fig. 4 BLI of Ca$^{2+}$ dynamics in the brain in awake mice. a** Illustration of intracranial viral administration and BLI of head-fixed awake mice. **b** Footshock-induced Ca$^{2+}$ in the basolateral amygdala (BLA). Top: Viral injection sites and the general experimental procedure. Middle: Bioluminescence intensity heatmap of a representative BRIC- or BREP-expressing mouse in response to 13 consecutive trials of footshock stimulations. Bottom: Quantification of intensity changes presented as mean ± s.e.m. ($n = 4$ mice, each with 13 trials). The right panel compares the average responses of 13 trials for each mouse. The $P$ value was derived from paired two-tailed $t$-tests. **c** Ca$^{2+}$ in the hippocampus (HPC) during kainic acid (KA)-induced seizures. Top: Viral injection sites and the general experimental procedure. Middle: Intensity traces of BRIC- or BREP-expressing mice. Bottom: Comparison of maximal intensity changes and area under the curve (AUC) presented as mean ± s.e.m. ($n = 6$ mice). $P$ values were derived from two-way ANOVA followed by Šídák's multiple comparisons test or unpaired two-tailed $t$-tests. The baselines for all intensity traces were corrected using a monoexponential decay model. BL, bioluminescence. Arb. units, arbitrary units. Source data are provided as a Source Data file.

BRIC was less prone to aggregation. When applied in mice, the BRIC signals in the presence of ETZ were much brighter and lasted much longer than OCaMBI110 with either FRZ or FFz. The integration of BRIC with ETZ allowed us to image Ca$^{2+}$ dynamics in awake mice minimally invasively with excellent signal-to-background and subsecond temporal resolution. However, despite the enhancement, BRIC is still suboptimal since its response to physiological Ca$^{2+}$ changes (e.g., from ~50 nM to ~1 µM) is limited to ~2-fold. We plan to further optimize the practical dynamic range of this indicator in future.

Our current approach macroscopically records concerted changes in a neuronal population that expresses the BRIC indicator. The spatial resolution is limited by scattering when photons

travel through brain tissue, skull, and skin. Also, photons reaching the detector are still scarce; when using the experiment setups presented here, the temporal resolution is in the second and subsecond ranges. Although the overall duration of the rising and extinction of intracellular Ca$^{2+}$ signals after the action potential is likely within this time range, the kinetics of Ca$^{2+}$ concentration changes is much faster than the achieved temporal resolution[56]. Thus, interested users should be aware that the recorded signals are undersampled over time. Considering these reasons and the fact that fluorescent Ca$^{2+}$ indicators, such as GCaMPs, have been extensively optimized[56], fluorescence remains the choice of imaging modality for fiber photometry or single-cell-resolution recording through cranial windows. On the

other hand, the method presented here offers technical simplicity, reduced invasiveness, and deep-brain imaging capability. The approach complements, but cannot replace, fluorescence or electrical recording.

This study tested ETZ in both BALB/cJ and C57BL/6 J strains of mice and used intracranial injection to deliver viral vectors. Recently, AAV vectors with engineered capsids have been reported to cross the BBB of C57BL/6 J mice and marmoset[57,58]. We are currently exploring these AAV vectors for peripheral gene delivery and transduction of specific neuronal populations in transgenic Cre mice with the C57BL/6 genetic background. We expect the effort to lead to an utterly noninvasive strategy for recording the activity of neuronal ensembles in small mammals.

Also, we had to apply baseline corrections for time-lapse intensity quantitation. A monoexponential decay model was adequate for baseline corrections in those experiments spanning short periods, although a more complex decay was evident during the more extended 60-min period shown in Fig. 3c. Further, BRIC is an intensiometric indicator, and we used head-fixed awake mice for macroscopic BLI. Unavoidable motions or blood flow changes in animals may cause intensity instability, resulting in potential artifacts when recording small $Ca^{2+}$ changes. Future developments may allow two-channel BLI of the mouse brain with high sensitivity, enabling ratiometric correction for baseline decays, blood flow changes, and animal motions.

In addition, a recent study described the use of FRZ and a NanoLuc-derived bioluminescent voltage indicator for imaging cortical activity in mice[31]. To gain enough signals, a cranial window was used for bioluminescence collection and continuous substrate delivery to the cortex. Furthermore, a GaAsP image intensifier was placed in front of the EMCCD to boost signals. Another recent preprint described a NanoLuc-derived bioluminescent glutamate indicator that has been tested in cultured cells[59]. We expect our results and strategies presented here to assist in the further development of these bioluminescent indicators for deep-brain in vivo imaging with minimally invasive procedures. Furthermore, as much-enhanced BLI tools, ETZ, BREP, and BRIC are expected to find diverse applications and generate broad impacts beyond neurobiology.

## Methods

**Ethical statement**. The study complies with all relevant ethical regulations, and all animal experiments were conducted according to the approval (Protocol #4196) and guidelines of the University of Virginia Institutional Animal Care and Use Committee. BALB/cJ mice (#000651) and C57BL/6 J mice (#000664) were purchased from the Jackson Laboratory and bred under standard conditions. Mice were hosted in a temperature-controlled room (~23 °C) with a 12 h/12 h dark-light cycle and ~50% humidity. Animals were randomly allocated to experimental groups with a balance of females and males at the age of ~8 weeks.

**Reagents and general methods**. Unless otherwise stated, all chemicals were purchased from MilliporeSigma, Fisher Scientific, or VWR and used without further purification. Kainic acid monohydrate (KA) was purchased from Cayman Chemical. pcDNA3.1-Orange_CaMBI_110 (Addgene #124094) and pAAV-hSyn.iGluSnFr.WPRE.SV40 (Addgene #98929) were gifts from M.Z. Lin and L.L. Looger, respectively. pAdDeltaF6 (Addgene #112867), pAAV2/9n (Addgene #112865) were gifts from J.M. Wilson. Synthetic DNA oligonucleotides (Supplementary Table 1) were purchased from Integrated DNA Technologies or Eurofins Genomics. Restriction endonucleases and Phusion High-Fidelity DNA Polymerase were purchased from Thermo Scientific. Products of PCR and restriction digestion were purified with preparative agarose gel electrophoresis, followed by gel extraction. DNA sequences were analyzed by Eurofins Genomics. Gibson assembly was performed using a homemade kit by following a procedure from Addgene. Small-scale plasmid DNA preparation was performed using Miniprep kits from Syd Labs. Large-scale plasmid DNA preparation was performed using alkaline lysis followed by isopropanol precipitation, PEG 8000 precipitation, and phenol/chloroform extraction. Merck Geduran Si 60 silica gel was used for normal-phase column chromatography. MassLynx (Version 4.2) was used to run a Waters Prep 150/SQ Detector 2 LC-MS Purification System equipped with an XBridge BEH Amide/Phenyl OBD Prep Column (130 Å, 5 µm, 30 mm × 150 mm) for preparative reverse-phase HPLC purification. Lyophilization was performed on a 12-port

Labconco freeze dryer with an Edwards RV3 vacuum pump. All NMR spectra were collected on a Bruker Avance III 600 MHz NMR spectrometer with the Bruker TopSpin IconNMR (Version 3.5pl4) software, and data were further analyzed using MestReNova (Version 12.0.3). Reference values for residual solvents were taken as 7.27 ($CDCl_3$) or 3.30 (methanol-$d_4$) ppm for $^1$H-NMR, and 77.2 ($CDCl_3$) or 49.0 (methanol-$d_4$) ppm for $^{13}$C-NMR. Splitting patterns of NMR active peaks are reported as s (singlet), d (doublet), t (triplet), dd (doublet of doublets), dt (doublet of triplets), and m (multiplet). The BMG Labtech CLARIOstar Plus Reader Software (Version 5.70 R2) was used to control the CLARIOstar Plus microplate reader, and results were automatically exported to the BMG Labtech MARS Data Analysis Software (Version 3. 42 R5) for analysis. Agilent MassHunter Workstation Data Acquisition (Version B.09) was used to acquire high-resolution mass spectra (HR-MS) on an Agilent 6545 Q-TOF LC/MS system via direct infusion. Agilent MassHunter Quantitative Analysis Navigator (Version B.08) was used to analyze HR-MS data. The UVP VisionWorksLS software (Version 8.6) and µManager (Version 2) were used to control a UVP BioSpectrum dark box and cameras for macroscopic BLI, while Leica LAS X (Version 3.5.7) or MicroManager (Version 2.0) was used to acquire microscopic images. The ChemAxon LogD Predictor (https://disco.chemaxon.com/calculators/demo/plugins/logd/) was used to predict the logD values of the specified compounds at pH 7.

**General information for chemical synthesis**. Synthetic schemes and compound numbering information are shown in Supplementary Fig. 2. NMR and HR-MS spectra for key compounds are presented in Supplementary Figs. 15–18. The detailed procedures for compound synthesis are described below. Furimazine (FRZ) and FFz were synthesized following reported methods[14,17].

**1,1-Diethoxy-3-phenylacetone (Compound 1)**. Compound **1** was synthesized by modifying a previous procedure[60]. Ethyl diethoxyacetate (10 mL, 56.8 mmol, 1.0 equiv.) was added into 30 mL anhydrous THF under argon (Ar) protection. Subsequently, BnMgCl (45 mL, 1.4 M in THF, 1.1 equiv.) was dropwise added into the reaction mixture in a dry ice-acetone bath. The mixture was stirred for 1 h in the dry ice-acetone bath, and then at room temperature for another 2 h. The reaction was next quenched by addition of $H_2O$. The mixture was filtrated and extracted three times with EtOAc. The organic layer was combined, dried with anhydrous $Na_2SO_4$, concentrated in vacuo. Purification was performed with silica gel chromatography (EtOAc/hexane = 1/5, v/v) to give compound **1** (8.8 g, 70.0 %).

**4-Oxo-4-(prop-2-yn-1-yloxy) butanoic acid (Compound 2)**. NHS (920.0 mg, 8.0 mmol, 0.4 equiv.), DMAP (260.0 mg, 2.0 mmol, 0.1 equiv.), TEA (840.0 µL, 6.0 mmol, 0.3 equiv.) and propargyl alcohol (7.4 mL, 60.0 mmol, 3.0 equiv.) were separately added into a solution of succinic anhydride (2000.0 mg, 20.0 mmol, 1.0 equiv.) dissolved in 50 mL anhydrous toluene. The mixture was heated to reflux overnight under Ar. After cooling, the mixture was washed with brine and extracted three times with EtOAc. The organic layer was combined and dried with anhydrous $Na_2SO_4$. After filtration and rotovap concentration, the residue was further purified with silica gel chromatography (EtOAc/hexane/acetic acid = 1/4/0.005, v/v/v) to afford compound **2** (2028.0 mg, 65%).

**2-Amino-3,5-diphenylpyrazine (Compound 3)**. Compound **3** was synthesized by modifying a previous method[15]. 2-Amino-3,5-dibromopyrazine (3000.0 mg, 12.0 mmol, 1.0 equiv.), phenylboronic acid (5800.0 mg, 48.0 mmol, 4.0 equiv.) and bis(benzonitrile)dichloro palladium (1700.0 mg, 2.4 mmol, 0.2 equiv.) were dissolved in EtOH. Next, 1 N $Na_2CO_3$ (48 mL, 4.0 equiv.) was added into the reaction mixture, which was further heated to reflux overnight under Ar. After cooling down to room temperature, the mixture was filtered, and EtOH was removed with rotovap. The residue was acidified to pH 4~5 with HCl (1 N) and washed three times with EtOAc. The aqueous layer was next alkalized to pH ~11 with NaOH (1 N) and extracted three times with EtOAc. The organic layer was combined, washed with brine and $H_2O$ three times, dried with anhydrous $Na_2SO_4$, filtered, and concentrated in vacuo to give compound **3** (2075.0 mg, 70%).

**Diphenylterazine (DTZ)**. DTZ was synthesized by modifying a previous method[15]. 6 N HCl (5.4 mL, 50.0 mmol.) was added to 5 mL of 1,4-dioxane in a 60-mL Ace pressure tube (Sigma-Aldrich #Z568767) containing compound **3** (150.0 mg, 0.61 mmol, 1.0 equiv.) and 1,1-diethoxy-3-phenylacetone (534.0 mg, 2.4 mmol, 4.0 equiv.). The tube was sealed, and the mixture was maintained at 120 °C with stirring overnight. Next, the mixture was cooled down to room temperature before the solvent was removed under reduced pressure. The crude was re-dissolved in 15 mL MeOH, and next, purified with the Waters preparative RPLC-MS (acetonitrile/water = 30:70 to 98:2, 20 mL/min). Product fractions were combined and lyophilized to give DTZ (126.0 mg, 55%).

**2-Benzyl-6,8-diphenylimidazo[1,2-a]pyrazin-3-yl prop-2-yn-1-yl succinate (Compound 4)**. Compound **2** (83.0 mg, 0.53 mmol, 2.0 equiv.) and DCC (82.0 mg, 0.4 mmol, 1.5 equiv.) were placed in an oven-dried two-neck round-bottom flask purged with Ar three times. Anhydrous DCM (10 mL) mixed with TEA (80.0 µL, 0.53 mmol, 2.0 equiv.) and two drops of anhydrous DMF was injected into the

reaction system with stirring at room temperature over 15 min. Subsequently, DTZ (101.7 mg, 0.27 mmol, 1.0 equiv.) was quickly added into the reaction system through one neck of the flask. The system was flushed with Ar and the reaction mixture was stirred for additional 20~30 min. The progress of the reaction was monitored with TLC (hexane/EtOAc = 3:1). After completion of the reaction, the mixture was cooled down to −20 °C to precipitate the DCU by-product. After filtration and concentration *in vacuo*, the residue was further purified with silica gel chromatography (EtOAc/hexane = 1:5, v/v) to give compound 4 (109.0 mg, 80 %). [1]H NMR (600 MHz, CDCl₃) δ 8.89–8.84 (m, 2H), 8.24 (s, 1H), 8.16 (dd, $J$ = 8.4, 1.2 Hz, 2H), 7.60–7.55 (m, 2H), 7.54–7.46 (m, 3H), 7.42–7.37 (m, 1H), 7.36–7.29 (m, 4H), 7.26–7.22 (m, 1H), 4.75 (d, $J$ = 2.5 Hz, 2H), 4.24 (s, 2H), 2.79–2.74 (m, 2H), 2.73–2.64 (m, 2H), 2.46 (t, $J$ = 2.5 Hz, 1H). [13]C NMR (151 MHz, CDCl₃) δ 171.7, 169.0, 147.8, 138.5, 138.1, 136.8, 136.1, 135.3, 133.3, 130.4, 130.2, 129.8, 129.2, 128.8, 128.7, 128.5, 128.4, 128.3, 126.4, 126.3, 109.7, 77.2, 75.5, 75.3, 52.6, 34.3, 28.8, 28.1. ESI-MS (m/z): $[M + H]^+$ calcd. for $C_{32}H_{25}N_3O_4$, 516.19; found, 516.18.

## 2-(4-(((4-((2-Benzyl-6,8-diphenylimidazo[1,2-a]pyrazin-3-yl)oxy)−4-oxobutanoyl)oxy) methyl)−1H-1,2,3-triazol-1-yl)acetic acid (ETZ).

Compound 4 (83.0 mg, 0.53 mmol, 2.0 equiv.), CuI (44.0 mg, 0.23 mmol, 2.0 equiv.) and THPTA (25.0 mg, 0.23 mmol, 0.5 equiv.) were placed in a two-neck round-bottom flask purged with Ar three times. Azidoacetic acid (53.0 µL, 0.47 mmol, 4.0 equiv.) was dissolved in THF (10 mL) and injected into the reaction system. Next, ascorbic acid (41.0 mg, 0.23 mmol, 2.0 equiv.) dissolved in 2 mL ddH₂O was dropwise injected into the reaction mixture and stirred overnight at room temperature. After completion of the transformation, the mixture was filtered and the product was purified with the Waters preparative RPLC-MS (acetonitrile/water = 30:70 to 98:2, v/v, 20 mL/min,). Product fractions were combined and lyophilized to give ETZ (35.0 mg, 50%). [1]H NMR (600 MHz, methanol-d₄) δ 8.69–8.64 (m, 2H), 8.51 (s, 1H), 8.16 (dt, $J$ = 6.6, 1.3 Hz, 2H), 7.92 (s, 1H), 7.57–7.52 (m, 3H), 7.45 (t, $J$ = 7.8 Hz, 2H), 7.39–7.34 (m, 1H), 7.33–7.23 (m, 4H), 7.22–7.17 (m, 1H), 5.28 (s, 2H), 5.15 (s, 2H), 4.14 (s, 2H), 2.87–2.84 (m, 2H), 2.81–2.77 (m, 2H). [13]C NMR (151 MHz, methanol-d₄) δ 178.2, 172.6, 169.7, 147.6, 142.3, 138.5, 138.1, 136.5, 135.9, 135.5, 132.9, 130.0, 129.4, 128.9, 128.7, 128.4, 128.1, 127.9, 126.1, 126.0, 125.8, 124.9, 110.1, 57.4, 50.4, 32.8, 28.3, 27.9. ESI-HRMS (m/z): $[M + H]^+$ calcd. for $C_{34}H_{28}N_6O_6$, 617.2149; found, 617.2149.

## 4-(2-(Dimethylamino)ethoxy)−4-oxobutanoic acid (Compound 5).

NHS (920.0 mg, 8.0 mmol, 0.4 equiv.), DMAP (260.0 mg, 2.0 mmol, 0.1 equiv.), TEA (840.0 µL, 6.0 mmol, 0.3 equiv.) and 2-dimethylaminoethanol (6.5 mL, 60.0 mmol, 3.0 equiv.) were consequently added into a solution of succinic anhydride (2000 mg, 20 mmol, 1.0 equiv) dissolved in anhydrous toluene (50 mL). The mixture was heated to reflux overnight under Ar. After cooling to room temperature, the solvent was removed *in vacuo*. The residue was dissolved in MeOH: ddH₂O (1:1, v/v) and purified with the Waters preparative RPLC-MS (acetonitrile/water = 3:97 to 98:2, v/v, 20 mL/min,). Product fractions were combined and lyophilized to give crude compound 5 (1580.0 mg, 42%).

## 2-Benzyl-6,8-diphenylimidazo[1,2-a]pyrazin-3-yl (2-(dimethylamino)ethyl) succinate (C3-DMA-DTZ).

Compound 5 (240.0 mg, 1.3 mmol, 4.0 equiv.) and DCC (132.0 mg, 0.64 mmol, 4.0 equiv.) were placed in an oven-dried two-neck round-bottom flask purged with Ar three times. Anhydrous DCM (10 mL), DMF (1 mL), and TEA (88.0 µL, 0.64 mmol, 2.0 equiv.) were immediately added into the reaction mixture with stirring at room temperature over 30 min. Subsequently, DTZ (122.0 mg, 0.33 mmol, 1.0 equiv.) was added to the reaction mixture. The resulting mixture was stirred for additional 80 min, then filtered, concentrated, and purified with silica gel chromatography (MeOH/DCM = 1/10, v/v) to give C3-DMA-DTZ (25.1 mg, 15 %). [1]H NMR (600 MHz, methanol-d₄) δ 8.68–8.64 (m, 2H), 8.52 (s, 1H), 8.18–8.14 (m, 2H), 7.58–7.51 (m, 3H), 7.48–7.44 (m, 2H), 7.40–7.36 (m, 1H), 7.32–7.26 (m, 4H), 7.23–7.18 (m, 1H), 4.28 (dd, $J$ = 6.0, 5.2 Hz, 2H), 4.16 (s, 2H), 2.88–2.84 (m, 2H), 2.83–2.78 (m, 2H), 2.66–2.62 (m, 2H), 2.27 (s, 6H). [13]C NMR (151 MHz, methanol-d₄) δ 173.2, 172.5, 130.0, 129.8, 129.4, 129.3, 128.7, 128.5, 128.4, 128.3, 128.2, 128.1, 128.0, 127.9, 127.8, 127.7, 126.2, 126.1, 126.0, 125.6, 125.5, 125.2, 58.2, 56.9, 43.9, 42.7, 32.1, 29.3, 28.2. ESI-HRMS (m/z): $[M + H]^+$ calcd. for $C_{33}H_{32}N_4O_4$, 549.2502; found, 549.2496.

## BREP library construction and screening.

To create genetic fusion libraries of mScarlet-I and teLuc, pcDNA3-LumiScarlet[15] and pcDNA3-teLuc-myc[16] were used as separate PCR templates. The mScarlet-I fragment was amplified from pcDNA3-LumiScarlet with the forward primer pBAD_FW_EP and one of the three reverse primers—pBAD_RV_BREP1_NNK1, pBAD_RV_BREP1_NNK2, and pBAD_RV_BREP1_3NDT (Supplementary Table 1). The teLuc fragment was amplified from pcDNA3-teLuc-myc with the reverse primer pBad_RV_EP and one of the three forward primers—pBAD_FW_BREP1_NNK1, pBAD_FW_BREP1_NNK2, and pBAD_FW_BREP1_3DTZ. The amplified mScarlet-I and teLuc fragments were used for three-part Gibson assembly along with pBAD/HisB predigested with Xho I and Hind III. The resultant DNA libraries were used to transform electrocompetent E. cloni 10 G cells (Lucigen), which were next allowed to grow on 2 × YT agar plates containing 100 µg/mL ampicillin and 0.02% (w/v)

L-arabinose at 37 °C overnight. 50 µM DTZ was sprayed onto colonies, and plates were imaged in a BLI system consisting of a UVP BioSpectrum dark box, a Computar Motorized ZOOM lens (M6Z1212MP3) and a QSI 628 Cooled CCD camera. Bioluminescence images were acquired with the µManager software. The brightest colonies were selected and cultured in 500 µL 2 × YT broth supplemented with 100 µg/mL ampicillin and 0.2% (w/v) L-arabinose in 96-well bacterial culture plates. After being shaken at 30 °C, 250 rpm for 24 h, cells were pelleted by centrifugation and further lysed with 500 µL of B-PER Bacterial Protein Extraction Reagents (Pierce) at 4 °C for 30 min. Cell lysates were prepared after centrifugation. 5 µL of each lysate was further diluted with 95 µL in vitro assay buffer (1 mM CDTA, 0.05% Tergitol NP-40, 0.05% Antifoam 204, 150 mM KCl, 100 mM MES, pH 6.0, 1 mM DTT, and 35 mM thiourea). Next, bioluminescence spectra were recorded on a BMG Labtech microplate reader with the equipped red-sensitive PMT right after injecting 100 µL of 25 µM DTZ pre-dissolved in the assay buffer mentioned above. Mutants with high BRET efficiency and brightness were selected.

## BRIC library construction and screening.

The CaM-M13 fragment was PCR-amplified from pcDNA3.1-Orange_CaMBI_110 with oligos pBad_FW2_CaM and pBad_RV2_M13. Next, pBAD-BREP was amplified with either pBad_FW1_mScarlet and pBad_RV1_teLuc(133), or pBad_FW3_teLuc(134) and pBad_RV3_te-Luc(168) to generate two fragments. The three fragments, along with pBAD/HisB predigested with Xho I and Hind III, were assembled in a four-part Gibson assembly reaction. The product was used to transform E. coli DH10B cells (Thermo Fisher). Cells were allowed to grow on 2 × YT agar plates supplemented with 100 µg/mL ampicillin and 0.02% (w/v) L-arabinose at 37 °C overnight. Single colonies were selected for growth in 2×YT liquid culture. Plasmids were recovered and Sanger sequencing confirmed the successful creation of pBAD-BRIC0.1. Next, Taq DNA polymerase (New England Biolabs) in the presence of MnCl₂ (0.05 mM) or GeneMorph II Random Mutagenesis Kit (Agilent Technologies) was used for error-prone PCR (EP-PCR)-based random mutagenesis. Oligos pBAD_FW_EP and pBAD_RV_EP were used for these reactions. PCR products were inserted into pBAD/HisB between Xho I and Hind III via Gibson assembly. The DNA libraries were used to transform E. coli DH10B cells, which were cultured on 2 × YT agar plates supplemented with 100 µg/mL ampicillin and 0.02% (w/v) L-arabinose at 37 °C overnight. Red colonies were selected and used to inoculate cultures in 96-well deep-well bacterial culture plates. Each well was filled with 1 mL of 2 × YT broth supplemented with 100 µg/mL ampicillin and 0.2% (w/v) L-arabinose. Cells were grown at 30 °C, 250 rpm for 48 h, pelleted by centrifugation, and lysed with 300 µL of B-PER at 4 °C for 30 min. After centrifugation, cell lysates were prepared. 5 µL of the supernatant was diluted with 185 µL of a Ca²⁺-free buffer (30 mM MOPS, 100 mM KCl, 10 mM EGTA, pH 7.2) and a Ca²⁺-containing buffer (30 mM MOPS, 100 mM KCl, 10 mM CaEGTA, pH 7.2) expected to provide 39 µM free Ca²⁺. Next, DTZ was dissolved in the in vitro assay buffer described above to the concentration of 500 µM, and 10 µL of the DTZ solution was injected into each well via a reagent injector in a BMG Labtech CLARIOstar Plus microplate reader. The final concentration of DTZ was thus 25 µM. The bioluminescence spectrum and intensity of each well were measured from 400 to 700 nm with 5-nm intervals. Mutants with high brightness and Ca²⁺ responsiveness were selected.

## BREP expression, purification, and emission spectrum recording.

BREP was expressed and purified following a previous procedure[16]. The purified protein was diluted with the in vitro assay buffer to a final concentration of 10 nM. 100 µL of the diluted protein was added into each well of a 96-well plate. 100 µL of 25 µM DTZ pre-dissolved in the in vitro assay buffer was injected into the well via a reagent injector in a BMG Labtech CLARIOstar Plus microplate reader. The mixture was shaken for 2 s before the bioluminescence spectrum was recorded using the equipped red-sensitive PMT. The instrument was set to scan from 400 to 700 nm with 5-nm intervals. Three technical repeats were performed to derive the average spectrum.

## BRIC and OCaMB110 expression and purification.

pBAD-BRIC was used to transform E. coli DH10B cells, which were next cultured on 2 × YT agar plates supplemented with ampicillin (100 µg/ml). A single colony was selected and cultured in 5 mL 2 × YT broth supplemented with 100 µg/mL ampicillin. The culture was shaken at 250 rpm and 37 °C overnight. The culture was next diluted with 500 mL of 2 × YT medium containing 100 µg/mL ampicillin. After shaking incubation at 37 °C for 2 h, protein expression was induced by adding L-arabinose (0.2%, w/v), and the culture was maintained at 250 rpm and 16 °C for 96 h. Cells were pelleted by centrifugation and resuspended in 10 mL of 1× phosphate-buffered saline (PBS, pH 7.4) supplemented with a cOmplet Mini EDTA-free Protease Inhibitor tablet (Roche). Cells were lysed by sonication, and the lysate was clarified by centrifugation at 15,000 × g for 30 min at 4 °C. The His₆-tagged protein was enriched with and then eluted from Ni-NTA agarose beads (Genesee Scientific). Finally, the protein was subjected to a size-exclusion HiLoad 16/600 Superdex 200 pg column (Cytiva) and eluted with an aqueous buffer containing 150 mM NaCl and 30 mM Tris HCl, pH 7.4.

A pBAD-OCaMB110 plasmid was created by amplifying the indicator gene from pcDNA3.1-Orange_CaMBI_110, but it turned to be challenging to prepare

high-purity OCaMB110 protein due to proteolysis. To address the issue, an additional Strep-tag II sequence was appended to the C-terminus of the reading frame, resulting in a pBAD-OCaMB110-Step plasmid. Protein expression was performed with the identical procedure described above. To purify the protein, the N-terminal His$_6$-tagged protein in cell lysates was first enriched with Ni-NTA agarose beads, and next, the eluate was applied to a Strep-Tactin Superflow high-capacity column (IBA Lifesciences). Finally, the eluate from the Strep-Tactin column was subjected to size-exclusion chromatography. A similar pBAD-BRIC-Step plasmid was constructed in parallel. The spectral properties of the BRIC protein prepared from pBAD-BRIC-Step with dual His$_6$ and Strep tags were identical to the protein prepared from pBAD-BRIC only with the N-terminal His$_6$ tag.

**Spectroscopic characterization of BRIC and OCaMB110.** Protein concentrations were determined with the Pierce 660 nm Protein assay using bovine serum albumin (BSA) standards. The Ca$^{2+}$-containing and Ca$^{2+}$-free buffers mentioned above were used to record the emission spectra for Ca$^{2+}$-free and Ca$^{2+}$-saturated states. Final protein and substrate concentrations were 50 nM and 25 μM, respectively. The Ca$^{2+}$ affinity was determined as previously described using a series of buffers made from mixing the Ca$^{2+}$-containing and Ca$^{2+}$-free buffers[29,61]. The bioluminescence intensity at 595 nm recorded on a BMG Labtech CLARIOstar Plus microplate reader was plotted against the expected free Ca$^{2+}$ concentration in each buffer. Three technical repeats were performed, and the data was fit with a one-site binding model in GraphPad Prism. To determine pH sensitivity, the Ca$^{2+}$-containing and Ca$^{2+}$-free buffers were adjusted with HCl (12 M and 1 M) or KOH (4 M) to gain buffers with pH values ranging from 5.5 to 8.0. Bioluminescence intensity ratios at 595 nm in the presence and absence of Ca$^{2+}$ were plotted against the pH values.

**Evaluation of luciferase substrates in BREP-expressing HEK 293 T cells.** pBad-BREP was amplified with oligos pcDNA3_FW_HindIII and pcDNA3_RV_XhoI via PCRs. The fragment was inserted into pcDNA3 to afford pcDNA3-BREP. pcDNA3-BREP was next used to transfect HEK 293 T cells (purchased from ATCC, Cat. # CRL-3216) using a described procedure[16]. Cells were collected 20 h post-transfection, and resuspended in and diluted with 1×PBS (pH 7.2). ~7,000 cells in 100 μL of total volume were placed in individual wells in a 96-well plate. The compounds (DTZ, ETZ, and C3-DMA-DTZ) were dissolved in 1×PBS (pH 7.2) to gain 50 μM concentrations. Next, 100 μL of the substrates were added to BREP-expressing HEK 293 T cells in the 96-well plate. BLI was performed immediately with a UVP BioSpectrum dark box, a Computar Motorized ZOOM lens (M6Z1212MP3), and an Andor iXon Life 888 EMCCD camera. The camera was first set to the "Photon Counting" mode using OptAcquire pre-settings, and the gain was next adjusted down to 500. Other parameters were: camera binning 1 × 1; camera sensor temperature −70 °C; 100 ms exposure time with acquisitions every 10 s. The lens was controlled using the UVP VisionWorksLS software with aperture set to f12.8, zoom set to 32%, and focus set to 0%. The plate was placed 27 cm away from the front of the lens with no emission filter. Images were acquired with the μManager software and processed in the Fiji version of ImageJ 2.1. Image stacks were first subtracted for background by setting the rolling ball radius to 100 pixels. Next, the region of interest (ROI) was selected by circling cell locations, and the intensity value integrated over the ROI was extracted for further analysis. Data were plotted and statistical analysis was performed in GraphPad Prism. After the software-based background subtraction, the images were left with residual background, so the ROI was moved away from the cell locations to evaluate residual background, which was further used to subtract signals for calculating the integrated bioluminescence intensity (area under the curve).

**Preparation of Adeno-Associated Viruses (AAVs).** Indicator genes were amplified from their corresponding pcDNA3/3.1 plasmids and inserted into a pAAV-hSyn vector to generate pAAV-hSyn-BRIC, pAAV-hSyn-BREP, and pAAV-hSyn-OCaMBI110. Next, individual transfer plasmids, along with pAd-DeltaF6 and pAAV2/9n, were used to transfect HEK 293 T cells to pack viruses. A protocol by Rego et al.[62] was followed for viral packing and purification. Viral titers were determined with quantitative PCR (qPCR) based on SYBR green by following a protocol from Addgene. Typical AAV titers were 5 × 10$^{14}$ − 1 × 10$^{15}$ GC/mL. AAVs were aliquoted and stored at −80 °C for long-term stability.

**Evaluation of luciferase substrates for brain delivery in mice.** Cells were cultured, transfected, and collected as described in the above section. 3 μL of cells (~7,000 cells) were delivered into each side of the hippocampus (AP −1.7, ML ± 1.2 and DV −1.5)[63] of 8-week-old, anesthetized BALB/cJ mice via intracranial stereotactic injection at a flow rate of 200 nL min$^{-1}$. When the infusion was complete, the needle was kept in the brain for 5 min before being withdrawn. The wound was then sealed with surgical adhesive.

Meanwhile, the compounds DTZ, ETZ, and C3-DMA-DTZ were dissolved in aqueous buffers supplemented with 25% (w/v) 2-hydroxypropyl-β-cyclodextrin (HP-β-CD) and 20% (v/v) PEG-400. More specifically, a 5 mL injection buffer for DTZ was made by dissolving 1.25 g HP-β-BRIC and 1 mL PEG-400 in ~3 mL normal saline; a 5 mL injection buffer for ETZ was made by dissolving 1.25 g HP-β-CD and

1 mL PEG-400 in ~3 mL of normal saline pre-supplemented with 5% (w/v) NaHCO$_3$; a 5 mL injection buffer for C3-DMA-DTZ was made by dissolving 1.25 g HP-β-CD and 1 mL PEG-400 in ~3 mL of normal saline pre-supplemented with 100 mM citric acid (pH 5.5). NaHCO$_3$ and citric acid have been used broadly for intravenous infusion and were supplemented to formulate ETZ and C3-DMA-DTZ as salts in situ. This buffer system was modified from the previously used recipe[15,19] with remarkably reduced viscosity, allowing more consistent intravenous injection.

100 μL of DTZ (2.5 mM), ETZ (6.8 mM), or C3-DMA-DTZ (4.5 mM) in their corresponding in vivo injection buffers was delivered to mice via tail vein right after intracranial cell injection. Bioluminescence images were acquired with a BLI system consisting of a UVP BioSpectrum dark box, a Computar Motorized ZOOM lens (M6Z1212MP3) and a QSI 628 Cooled CCD camera. The settings were: camera binning 4 × 4; high gain; camera sensor temperature −15 °C; 10 s exposure time and each acquisition every 60 s. The lens was controlled using the UVP VisionWorksLS software with aperture set to 100% open, zoom set to 0%, and focus set to 0%. Anesthetized mice were placed 21 cm away from the front of the lens with no emission filter used. To minimize biological variables, each mouse was sequentially tested with three substrates, and the next substrate was administered after the signal of the previous substrate faded out. A total of three mice were used, and the substrate injection order was rotated for each mouse to control the bias.

To further evaluate the compounds with endogenous neurons in mice, 500 nL of BREP AAV (~1 × 10$^{15}$ GC/mL) was delivered to each side of the hippocampus of 8-week-old BALB/cJ mice via intracranial stereotactic injection at a flow rate of 100 nL min$^{-1}$ using the coordinate described above. Three weeks later, mice were used for brightness comparison. ETZ (6.8 mM), ETZ (2.5 mM), DTZ (2.5 mM), C3-DMA-DTZ (4.5 mM), FFz (6.0 mM), and FRZ (3.0 mM) were dissolved in their corresponding in vivo injection buffers, and 100 μL of each substrate was delivered to mice via tail vein. BLI was subsequently performed with a UVP BioSpectrum dark box, a Computar Motorized ZOOM lens (M6Z1212MP3), and an Andor iXon Life 888 EMCCD camera. The camera was first set to the "Photon Counting" mode using OptAcquire pre-settings, and the gain was next adjusted down to 500. Other parameters were: camera binning 2 × 2; camera temperature −70 °C; 500 ms exposure time with acquisitions every 10 s. The lens was controlled using the UVP VisionWorksLS software with aperture set to 100% open, zoom set to 0%, and focus set to 0%. Anesthetized mice were placed 27 cm away from the front of the lens with no emission filter used.

Images were acquired with the μManager software and processed in the Fiji version of ImageJ 2.1. Image stacks were first subtracted for background by setting the rolling ball radius to 100 pixels. Next, the region of interest (ROI) was selected based on bioluminescence from the mouse brain, and the intensity value integrated over the ROI was extracted for further analysis. Data were plotted and statistical analysis was performed in GraphPad Prism. After the software-based background subtraction, the images were left with residual background, so the ROI was moved away from the mouse brain region to evaluate residual background, which was further used to subtract signals for calculating the integrated bioluminescence intensity (area under the curve).

**Characterization and comparison of BRIC and OCaMBI110 in HeLa cells.** pBad-BRIC was used for PCRs with oligos pcDNA3_FW_HindIII and pcDNA3_RV_XhoI. The fragment was inserted into pcDNA3 to afford pcDNA3-BRIC. HeLa cells (purchased from ATCC, Cat. # CCL-2) were transfected with 3 μg of the plasmid pcDNA3-BRIC or pcDNA3.1-Orange_CaMBI_110 using a described procedure[16]. Cells were allowed to grow at 37 °C in a 5% CO2 incubator for 18 h. Cells were rinsed twice with 1 × PBS and placed in 1 × PBS 15 min before imaging. Images were acquired on an inverted Leica DMi8 microscope equipped with a Photometrics Prime 95B Scientific CMOS camera. 40 μM DTZ or 40 μM FRZ was supplemented for bioluminescence. Imaging settings were: 40× oil immersion objective lens (NA 1.2), no filter cube, 2 × 2 camera binning, 1 s exposure with 0 s interval, camera sensor temperature −20 °C; 12-bit and high sensitivity mode. Histamine was dissolved in 1 × PBS and added during time-lapse imaging to a final concentration of 100 μM. Image stacks were processed as described in the previous section, except that ROIs were selected for individual cells and the mean of intensity values over the ROI was extracted for further analysis. Moreover, the baselines caused by substrate decay were fitted to a monoexponential model: $Y = (Y_0\text{-Plateau})*\exp(-K*X) + \text{Plateau}$. Data were plotted and statistical analysis was performed in GraphPad Prism.

**Characterization and comparison of BRIC and OCaMBI110 in primary mouse neurons.** Freshly extracted embryonic day 18 primary mouse brain tissue was collected for neuron dissociation. The neurons were plated on poly-D-lysine coated 35 mm glass-bottom dishes with 2 mL NbActiv4 medium (BrainBits) at 37 °C and 5% CO$_2$. Half of the medium was changed to fresh NbActiv4 every 2 days. Neurons were transduced with AAVs (AAV-hSyn-BRIC, AAV-hSyn-BREP, and AAV-hSyn-OCaMBI110) at the 5th day post plating. 3 μL of each virus at ~5 × 10$^{14}$ GC/mL and 1 μL of 1 M HEPES (pH 7.4) were added to each 35 mm culture dish (with cells at ~ 60% confluency). Half of the medium was replaced with fresh NbActiv 24 h later and then every 2 days. Neurons were evaluated on the fourth or fifth day after AAV transduction. Because extensive fluorescent aggregations were observed in OCaMBI110-expressing neurons on day 5, neurons on day 4 after AAV

transduction were used for quantitative comparison. Growth medium was replaced with 0.8 mL of the luminescence imaging buffer[29] (0.49 mM $MgCl_2$, 2 mM $CaCl_2$, 0.4 mM $MgSO_4$, 0.44 mM $KH_2PO_4$, 5.3 mM KCl, 4.2 mM $NaHCO_3$, 0.34 mM $Na_2HPO_4$, 138 mM NaCl, 10 mM HEPES pH 7.2, 15 mM D-glucose, and 0.1 mM sodium pyruvate) supplemented with 100 μM of DTZ or FRZ. Images were acquired on an inverted Leica DMi8 microscope equipped with a Photometrics Prime 95B Scientific CMOS camera. During time-lapse imaging, 0.2 mL of the high $K^+$ stimulation buffer[29] (0.49 mM $MgCl_2$, 2 mM $CaCl_2$, 0.4 mM $MgSO_4$, 0.44 mM $KH_2PO_4$, 143.2 mM KCl, 4.2 mM $NaHCO_3$, 0.34 mM $Na_2HPO_4$, 10 mM HEPES pH 7.2, 15 mM D-glucose, and 0.1 mM sodium pyruvate) was added to depolarize cells. Instrumental settings and data analysis were identical to those described for the HeLa cell experiment, except that the exposure time was 2 s. To use BRIC to image ionomycin-induced $Ca^{2+}$ influx, neurons on day 4 after BRIC AAV transfection were placed in 1 mL of the luminescence imaging buffer supplemented with 100 μM DTZ. During time-lapse imaging, 1 μL of 10 mM ionomycin was added to gain a final concentration of ~10 μM. The luminescence imaging buffer described above contains 2 mM $Ca^{2+}$. In the experiment with 10 mM extracellular $Ca^{2+}$, the luminescence imaging buffer was additionally supplemented with 8 mM $CaCl_2$. Instrumental settings and data analysis were unchanged from the high $K^+$ depolarization experiment.

**Comparison of BRIC and OCaMBI110 brightness in the hippocampus in mice.** 500 nL AAV (~1 × 10^15 GC/mL) was delivered to each side of the hippocampus of 8-week-old BALB/cJ mice via intracranial stereotactic injection at a flow rate of 100 nL min^−1 using the coordinate described above. 19 days after viral injection, mice were utilized for brightness comparison. 100 μL ETZ (6.8 mM) in the in vivo injection buffer was intravenously delivered into the anesthetized BRIC-expressing mouse. Subsequently, BLI was performed with a UVP BioSpectrum dark box, a Computar Motorized ZOOM lens (M6Z1212MP3), and an Andor iXon Life 888 EMCCD camera. The camera was first set to the "Photon Counting" mode using OptAcquire pre-settings, and the gain was next adjusted down to 500. Other parameters were: camera binning 2 × 2; camera temperature −70 °C; 500 ms exposure time with acquisitions every 5 s. The lens was controlled using the UVP VisionWorksLS software with aperture set to 100% open, zoom set to 0%, and focus set to 0%. Anesthetized mice were placed 21 cm away from the front of the lens with no emission filter used. FRZ and FFz were dissolved in the in vivo injection buffer (the same recipe described above for dissolving DTZ). 100 μL of FRZ (3.0 mM) or FFz (6.0 mM) were delivered into each of the anesthetized OCaMBI11-expressing mice. Imaging conditions were unchanged. Data analysis (including residual background analysis) was identical to those described above for comparing the brain delivery of the substrates.

**BRIC responses in the acute hippocampal slices.** Acute brain slices were prepared 3 weeks post viral delivery, which is described in the previous section. Freshly extracted mouse brains were pre-cooled and sliced to be 350 μm thickness in an ice-cold ACSF buffer (2.5 mM KCl, 119 mM NaCl, 1.3 mM $MgSO_4$, 26 mM $NaHCO_3$, 1 mM $NaH_2PO_4$, 2 mM $CaCl_2$, and 10 mM glucose, 95% $O_2$/5% $CO_2$). Next, brain slices were recovered in ACSF at 37 °C for 30 min and next placed in 2.0 mL of the luminescence imaging buffer (described above for neuron experiments) supplemented with 100 μM of DTZ. Images were acquired on a Scientifica SliceScope Pro 1000 equipped with a Photometrics Prime 95B Scientific CMOS camera. Imaging settings were: 4× objective lens (NA 0.1), no filter cube, 2 × 2 camera binning, 2 s exposure with 0 s interval, camera sensor temperature −15 °C; 12-bit high sensitivity mode. During time-lapse imaging, 0.5 mL of the high-$K^+$ stimulation buffer (described above for neuron experiments) was slowly added using a perfusion pump at a rate of 11.9 μL s^−1. This ~ 42 s process minimized the motion of the brain slice. BREP-expressing slices were prepared and imaged using the same procedures. Image processing and data analysis were identical to those described for the HeLa cell experiment.

**In vivo imaging of brain activities in awake mice.** 500 nL of the virus was at a flow rate of 100 nL min^−1 delivered to each side of the basolateral amygdala (BLA) of 8-week-old BALB/cJ mice and each side of the hippocampus of 8-week-old C57BL/6 J mice via intracranial stereotactic injection. The coordinate for BLA was: from Bregma, AP −1.42, ML ± 3, DV −5.6[64]. The same coordinate described above was used for the hippocampus. The viral titers for BREP and BRIC were ~5 × 10^14 and ~1 × 10^15 GC/mL, respectively. BLI was performed two to four weeks later. For $Ca^{2+}$ dynamics in the BLA included by footshock stimuli, 100 μL ETZ (6.8 mM) in the in vivo injection buffer was administered to an awake mouse via tail vein. The animal was next mounted on a Narishige plastic mouse head holder (SRP-AM2). Subsequently, BLI was performed with a UVP BioSpectrum dark box, a Computar Motorized ZOOM lens (M6Z1212MP3), and an Andor iXon Life 888 EMCCD camera. Instrumental settings were identical to the descriptions for in vivo BRIC and OCaMBI110 brightness comparison, except that the binning was 4 × 4 and the exposure was 1 s. Mice were placed 27 cm away from the front of the lens. Each experiment consisted of 100 s for animal acclimation, followed by 13 footshock trials. Each trial begins with a 0.8-mA electric footshock (lasting for 1 s), and the intervals between footshock stimuli were 40 s. The electric shock was generated

with an A-M Systems 2100 isolated pulse stimulator. For $Ca^{2+}$ dynamics in the hippocampus, KA was first delivered to mice via i.p. injection at a dosage of 20 mg per kg body weight. If no seizure was observed within 2 hours, a second dose KA (5 mg per kg body weight) was delivered via i.p. injection. Finally, when an evident tremor was observed, 100 μL ETZ (6.8 mM) in the in vivo injection buffer was administered to the mouse via tail vein. Next, BLI was performed with the head-fixed animal as described above. Data analysis was identical to those described for the HeLa cell experiment, except that ROIs were selected for bioluminescence from the brain.

**Evaluation of the background signal of ETZ in mice.** 100 μL ETZ (6.8 mM) in the in vivo injection buffers was delivered to 8-week-old, anesthetized BALB/cJ mice via tail vein. BLI was subsequently performed with a UVP BioSpectrum dark box, a Computar Motorized ZOOM lens (M6Z1212MP3), and an Andor iXon Life 888 EMCCD camera. Mice were placed 21 cm away from the front of the lens with no emission filter. Other imaging conditions and data analysis procedure were identical to the descriptions in the above section. The same procedure was used to examine mice with untransfected HEK 293 T cells intracranially injected into the hippocampus.

**General procedure to correct time-lapse images for baseline decays.** The Fiji version of the ImageJ 2.1 was used for image processing. Image stacks were first subtracted for background by setting the rolling ball radius to 100 pixels. Next, an average image of each stack was used to generate a binary mask, which was subsequently applied to the image stack. Thus, the information for pixels within the ROI was retained, and the intensity values for other background pixels were set to 0. This conversion was necessary so that the background signals were not amplified during the subsequent baseline decay correction. Next, the whole image stacks were subjected to "photobleaching correction". This procedure essentially fitted the image stacks with a monoexponential model: $Y = (Y_0\text{-Plateau}) \times \exp(-K \times X) + \text{Plateau}$, and the intensity of each image in the stack was rescaled. This correction procedure was adequate for histamine-induced $Ca^{2+}$ dynamics in HeLa and in vivo imaging of footshock- and KA-stimulated mice. The imaging data for high $K^+$-induced $Ca^{2+}$ in cultured neurons and brain slices required a more complicated correction. Because high $K^+$ induced a relatively large, concerted intensity increase, the baseline was not properly identified when the whole stack was used for monoexponential fitting. Instead, the mean intensity values of each image in the stack were extracted and exported to Microsoft Excel. Data points during the expected peak responses were excluded, and monoexponential fitting was applied to the remaining data points. Finally, the offline-derived decay parameters were used to correct the whole image stacks in Fiji.

**Statistics & reproducibility.** Fiji (ImageJ Version 2.1) was used to analyze microscopic images. Imaging background was typically subtracted by setting the rolling ball radius to 300 pixels and more detailed procedures for imaging processing are presented in the above sections. Microsoft Excel (Version 15.21.1), GraphPad Prism (Version 8), and Affinity Designer (Version 1.10.4) were used to analyze data and prepare figures for publication. Sample size and the number of replications for experiments are presented in figure legends. No statistical methods were used to pre-determine the sample size. No data exclusions were performed. Blinding was not implemented during sensor development and in vitro and cellular characterization experiments. For in vivo footshock and KA-induced seizure experiments, the investigator who acquired and analyzed data was blinded to the group allocation during experiments. Data are shown as mean and standard deviation (s.d.) or standard error (s.e.m.), and the information is included in figure legends. Sample size and statistical methods used to calculate $P$ values are provided in figures or figure legends, and default settings in GraphPad Prism were used for these calculations. The significant confidence interval was set at 95%. The exact $P$ values are presented in figures or figure legends, except when $P$ is less than 0.0001 because GraphPad Prism does not provide extract $P$ values below 0.0001.

**Reporting summary.** Further information on research design is available in the Nature Research Reporting Summary linked to this article.

## Data availability

The plasmids pcDNA3-BREP (#172337), pcDNA3-BRIC (#172338), pAAV-hSyn-BREP (#172340), pAAV-hSyn-BRIC (#172341), and pBAD-BRIC (#172343) and their sequence information have been deposited to Addgene. All key data and experimental methods are presented in the main text or the supplementary materials. Source data are provided with this paper. Protein structures (Entries 2BBM, 7MJB, and 5LK4) used for creating graphs can be accessed from the RSCB Protein Data Bank. Source data are provided with this paper.

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

## Acknowledgements

We thank Dr. Zefan Li for confirming the result presented in Fig. 2b, and other members in the Ai lab for helpful discussion and assistance with experiments. We also acknowledge the UVA NMR Spectroscopy Core Facility for technical assistance. Research reported in this publication was supported by the University of Virginia Start-up Fund and National Institutes of Health grants (R01DK122253, R01GM129291, and RF1AG077773) to HA.

## Author contributions

H.A. conceived the project. X.T. synthesized compounds, engineered and characterized BRIC in vitro and in cultured cells, and prepared virus. Y.Z. performed in vivo viral injection. X.L. prepared primary neurons. X.T., X.L. and Y.Z. performed in vivo brightness comparison of substrates and indicators. Y.Z. and X.T. imaged brain slices and live animals. T.W. prepared some brain slices and conducted intravenous injections of substrates. Y.X. developed and recorded the emission of BREP. X.T., Y.Z., X.L. and Y.X. analyzed data and prepared figures. H.A., X.T., Y.Z, and Y.X. wrote the manuscript.

## Competing interests

The authors declare no competing interests.

Although there is no current plan to patent ETZ and BRIC, HA was an inventor of a patent (US Application # 15/694238) about DTZ and teLuc awarded to the University of California. Also, the University of Virginia filed a patent application (US Application # 17/434351) covering BREP, and HA and YX were listed as inventors. The remaining authors declare no competing interests.
