## [Peer Review File · Nature Communications]

Reviewers' Comments:

Reviewer #1:

Remarks to the Author:

Ai and coworkers report advances in the ability to detect brain calcium using bioluminescence. They describe a brighter, better-behaved NanoLuc-based calcium sensor, and a luciferin prosubstrate to address difficulty in delivering luciferase substrates into the brain. Together, this approach allowed imaging of brain calcium in awake mice. Overall, these advances should be of broad interest and worthy of publication after addressing the following comments:

The performance of the prosubstrate ETZ is intriguing but could use some clarification. Similar esterase-labile prosubstrates of coelenterazine such as EnduRen and ViviRen have been known for a long time and should be referenced (Promega). A prosubstrate strategy for delivering firefly luciferins into the brain has been previously reported using FAAH, an amidase highly expressed in the brain (Mofford et al., 2015; Adams et al., 2016). What candidate esterases for ETZ are present in the brain that could account for the observed results? As the authors note, hydrolysis may occur before brain entry, and it would seem likely that substantial esterase activity capable of hydrolyzing ETZ and C3-DMA-DTZ would be associated with other tissues such as the liver. One concern is that the comparisons of DTZ, ETZ, and C3-DMA-DTZ reported in Fig 1 and Supp Fig 3 were all conducted in mice with implanted luciferase-expressing HEK293T cells rather than in the AAV-transduced animals which express the luciferase in mouse neurons. The specific esterase activity and uptake properties of exogenously implanted HEK293T cells may therefore play a role that differs from endogenous mouse neurons. Please check whether ETZ, C3-DMA-DTZ, and DTZ differ in their ability to deliver DTZ into HEK293T cells *in vitro* and also include a direct comparison of these same prosubstrates in mice transduced with the AAV-delivered hSyn1-luciferase reporters.

Part of the rationale given for including a carboxylate in ETZ is to prevent BBB exclusion by Pgp. Please include discussion of ABCG2, which also operates at the BBB but has different substrate specificity. Decreasing lipophilicity was also proposed to increase brain delivery. The computed logP of DTZ is given as 4.3, but the logP values for ETZ and C3-DMA-DTZ are not described, and C3-DMA-DTZ performs significantly worse than DTZ. Please include these logP values and discuss. Are the results consistent with the hypothesis about decreasing lipophilicity? Or does it suggest the importance of other factors such as the rate of hydrolysis, binding to serum proteins, artifacts arising from the use of HEK293T cells, or something else? Please include other prosubstrate examples (such as the readily available compound 4) to gain better insight into the factors involved.

Enhanced BRET efficiency in BRED was achieved by optimizing the linker between teLuc and mScarlet. Following the general strategy of Oh et al. for Orange CaMBIs, a Ca²⁺-sensitive reporter BRIC with greater emission >600 nm was engineered that was more responsive to Ca²⁺ than Orange CaMBIs in the authors' hands, but less responsive than what was previously reported by Oh et al. The discrepancy was reasonably explained as potentially arising from oligomerization of dimeric CyOFP1 in the Orange CaMBI reporter, and evidence is presented that BRIC is better behaved. AAV delivery into the brains of live mice was fruitfully applied to compare the performance of BRIC, BRED, and OCaMBI *in vivo*, but comparison of DTZ, ETZ, and C3-DMA-DTZ should also be performed in these mice.

Other comments

The new compounds, particularly the prosubstrates ETZ and C3-DMA-DTZ, should have HRMS exact mass data.

Ref 39 describes AAV-PHP.eB et al. that can cross the BBB in several mouse strains, but notably not the BALB/c strain used in this manuscript.

"We named this new compound ETZ for esterase-dependent activation and enhanced *in vivo* performance". Does the E stand for esterase or enhanced? Both are underlined.

"Photos" within the optical window should be "photons".

Reviewer #2:

Remarks to the Author:

The manuscript, "A luciferase prosubstrate and a red bioluminescent calcium indicator for imaging neuronal activity in awake mice" by Tian et al. introduces a new red-shifted calcium sensor system that is based on bioluminescence. By virtue of its emission profile, it appears to penetrate tissue readily and therefore sets the stage for non-invasive Ca^{++} flux imaging of neural activity in the brain. As part of their system, they also optimize the substrate for the new fusion bioluminescence reporter. They compare their system with that of the currently most red-shifted bioluminescence sensor of Ca^{++} , Orange CaMBI, and the new system appears to out-perform Orange CaMBI.

On the other hand, the brevity of the manuscript appears to correspond with significant missing information and/or important control experiments not shown. This makes the evaluation of the study rather difficult. For example:

1. the authors description of the advantages/disadvantages of aequorin is inaccurate. Aequorin doesn't emit photons slowly---in fact, when activated by Ca^{++} , aequorin and obelin are very fast and can track action potentials. The major problem with aequorin for sensing neural activity over time is that aequorin discharges when activated by Ca^{++} and it only slowly recharges in the presence of coelenterazine (also, it is blue emission which doesn't penetrate tissue well).
2. Fig. 1D: No images of background luminescence controls are shown for the injection of non-BRIC expressing HEK cells, followed by the administration of DTZ and ETZ. This is important, especially since it appears that the researchers are injecting a higher concentration of ETZ than DTZ. For example, if the researchers inject 100uL of equal concentrations of ETZ and DTZ (2.5mM each instead of 6.8 and 2.5mM), do they still see an improvement in bioluminescence? Is the main advantage of the new substrate that a higher concentration can be injected due to the increased solubility?
3. Fig. 2: BRIC response is only 2X from $10e7$ to $10e6$ Ca^{++} , which is not a large dynamic range. If true, the authors need to acknowledge this less-than-optimal characteristic. Also, since this is an INTENSITY reporter (i.e., not a ratiometric reporter), it will therefore be subject to movement artifacts in a freely moving animal. Please acknowledge these limitations.
4. Fig 2G: clarify that the measurement is of bioluminescence intensity (what wavelength range?) and not a ratiometric reading.
5. Supp. Fig. 3: add furimazine to this experiment for comparison. Even in the original paper (Yeh et al. 2017), I don't see a clear comparison of teLuc with furimazine.
6. Fig. 3: FFz is only tested with OCaMBI, but teLuc is also a NanoLuc variant. Therefore, BRIC needs to be tested with FFz too. Also, critical controls are missing: what is the result when animals that have not been injected with AAV/sensor are treated with the substrate? Is there really no background? Then show us!
7. Fig 4B: generally the foot shock data are very nice. Can the authors compare the kinetics of BRIC with those of GCaMP, which is the main competing technology? For example, the authors should inject GCaMP into the amygdala & perform fiber photometry w/ 500ms exposure with an EMCCD camera, and compare the width of the response to that of their BRIC sensor.
8. Fig 4B: what do the raw traces look like over the entire recording relative to the start of the substrate injection? From Fig 3C it seems like there is a fast decay in signal from the time of the injection, so it is important to determine how that affects the size of the responses to the foot shocks.
9. Fig 4C: why is the calcium activity so variable across the mice for seizure? Is this an issue with the assay for detecting calcium, or is it a problem with the substrate not working consistently

when injected?

10. Fig 4B/C: there appears to be a small (but significant) increase in activity in the calcium-independent BREP when the animals receive a foot shock in 4B, and when the animals are undergoing seizure in 4C. What is causing that? Motion? Increased blood flow due to neural activity? Does this limit the usefulness of the sensor system if there are potentially small calcium-independent changes in bioluminescence with strong neural activity/motion? Address these limitations in the Discussion.

Minor points:

Add Reference(s) to the Introduction to support the important points that BLI is superior to fluorescence in terms of photobleaching, phototoxicity, long-term measurements (e.g., circadian rhythms), and as optogenetic partners.

The manuscript would benefit from some minor editing of English to enhance clarity.

Reviewer #3:

Remarks to the Author:

This manuscript describes the engineering of a two-component system enabling basic bioluminescence imaging (BLI) of calcium activity in awake mouse brain. The system comprises a luciferin prosubstrate with more efficient delivery across blood-brain barrier (BBB) as well as red-shifted emission with derivatives of teLuc, a NanoLuc derivative previously engineered by some of the authors. The second component of the system is an improved calcium-sensitive luciferase derived from teLuc that has additionally been optimized in a FRET construct with an mScarlet acceptor, generating red-shifted emission useful for better penetration through tissue and less scattering in general.

Both of these products are of great interest to many in the community, where imaging neuronal activity in intact awake animals remains an extremely challenging technical hurdle. In my opinion, BLI will ultimately supplant fluorescence imaging in live animals, once all of its many limitations get worked out. This manuscript represents an important step towards this future by markedly improving brain delivery of luciferins and by improving the dynamic range and signal strength of BL calcium sensors.

The authors should be cautious about overstating the advantages of the described BRIC sensor. While certainly a notable improvement over previous generations of BL calcium sensor, BRIC is still suboptimal in several ways. While the in vitro characterization of the sensor shows a ~6-fold dynamic range, the practical dynamic range is quite a bit smaller, considering that the K_d for calcium binding is ~133nM and the Hill coefficient appears to be fairly low as well, and so the signal change within the physiological range (~50nM to ~1uM) is in fact considerably smaller than the full dynamic range. This is the likely reason that HeLa cells generated only ~2-fold changes in BL intensity with calcium oscillations.

When characterizing the BRIC sensor in cultured neurons, a more gentle stimulation than depolarization with high KCl would be a welcome addition to this manuscript. Ideally, titration of the sensor in living cells (with CaEGTA buffers and ionomycin, for example) would provide very useful information about the differences in sensor behavior in vitro vs. in cellulo. However, I accept the evidence shown in intact mice subjected to foot shock as a legitimate demonstration of the sensor's basic functionality.

In general, the authors have provided sufficient evidence for most claims made in this manuscript. However, it should be noted that while it appears superficially that KA-induced seizures do indeed lead sometimes to detectable changes in BL intensity from the BRIC sensor, the variance in these experiments was extremely high and the authors are clearly struggling to find a condition that reaches statistical significance. The data presented in Fig. 4C is unfortunately not entirely

convincing, since some BRIC animals showed essentially no response. This might be solved by adding more animals, but my suspicion is that this seizure model simply doesn't produce strong enough calcium signals over a large enough area to work with the sensor as it currently stands (likely because its K_d and Hill coefficient are both too low, or because the population of neurons generating high calcium signals is too small). I suggest removing this from the manuscript, as it is really not necessary to demonstrate the utility of the sensor in its current form. Alternatively, the authors could include further discussion about this experiment and why the results are less robust than those from other experiments in this study.

The authors should consider clarifying some of the limitations of the current system in terms of spatial and temporal resolution. A brief list of potential biological questions that could be addressed using the BRIC/ETZ system would enhance the manuscript quite a bit. It appears that exposure times for all imaging in this study are between 500ms and 2s, a fairly major limitation when many calcium signals occur on the order of tens of milliseconds. It's important to acknowledge these limitations in the discussion section at some point more clearly, if only to keep expectations realistic among potential end users of this technology. What varieties of brain activity occur on achievable time and spatial scales?

The methodology is sound, with sufficient descriptions in the methods section for others with expertise in the field to replicate. Data are presented in a clear and logical way throughout the manuscript. An additional supplementary figure including the chemical structures of other luciferins used or mentioned in the manuscript would be very helpful (e.g. furimazine and FFz along side DTZ, ETZ, C2-DMA-DTZ for comparison). There are minor typographical errors throughout that need to be corrected, but no major issues with readability or content of the manuscript.

We thank the reviewers for their generally positive comments. We also appreciate the constructive comments, which have helped up improve the paper significantly. We have now performed additional experiments and revised our manuscript accordingly. Below we cite the raised concerns and provide our point-by-point responses.

Editorial comments:

** To maximise the reproducibility of research data, we strongly encourage you to provide a file containing the raw data underlying the following types of display items The “Data Availability” section should also include the statement “Source data are provided with this paper.”*

Response: We now provide an Excel file named “source data”, including data for box/bar graphs, dot plots, and line graphs.

** Please replace your bar graphs with plots that feature information about the distribution of the underlying data. All data points should be shown for plots with a sample size less than 10. For larger sample sizes, please consider box-and-whisker or violin plots as alternatives. Measures of centrality, dispersion and/or error bars should be plotted and described in the figure legend.*

Response: We have modified our figures accordingly.

Reviewer 1

Review 1 commented that the study “*should be of broad interest and worthy of publication*”. A few relatively minor concerns were raised, which have been addressed as discussed below:

The performance of the prosubstrate ETZ is intriguing but could use some clarification. Similar esterase-labile prosubstrates of coelenterazine such as EnduRen and ViviRen have been known for a long time and should be referenced (Promega). A prosubstrate strategy for delivering firefly luciferins into the brain has been previously reported using FAAH, an amidase highly expressed in the brain (Mofford et al., 2015; Adams et al., 2016).

Response: Thanks for the suggestion. We have added the following discussion to the manuscript: “Of note, several other studies developed C-3 caged CTZ or FRZ analogs, some of which also slowed bioluminescence decays in living cells.^{50,51} Particularly, a substrate manufactured by Promega Corporation, namely ViviRen, was shown to improve the light output of *Renilla* luciferase in live mice bearing brain tumors, compared to the equimolar amount of CTZ.³⁰ However, these previously explored C-3 caging groups were typically hydrophobic, unfavorably reducing substrate solubility in aqueous solutions. Furthermore, a prosubstrate strategy was used to deliver the amide derivatives of firefly luciferins into the mouse brain for fatty acid amide hydrolase (FAAH)-dependent activation.^{52,53}”

What candidate esterases for ETZ are present in the brain that could account for the observed results? As the authors note, hydrolysis may occur before brain entry, and it would seem likely that substantial esterase activity capable of hydrolyzing ETZ and C3-DMA-DTZ would be associated with other tissues such as the liver.

Response: Carboxylesterases are likely responsible for the hydrolysis of ETZ. We have expanded the discussion: “When administered to live cells or mice, ETZ is expected to be hydrolyzed by esterase to liberate DTZ.^{47,48} Since significant carboxylesterase activities were detected in mouse liver and blood,^{47,49} the labile ester linkages of ETZ may break down before it crosses the BBB.

*One concern is that the comparisons of DTZ, ETZ, and C3-DMA-DTZ reported in Fig 1 and Supp Fig 3 were all conducted in mice with implanted luciferase-expressing HEK293T cells rather than in the AAV-transduced animals which express the luciferase in mouse neurons. The specific esterase activity and uptake properties of exogenously implanted HEK293T cells may therefore play a role that differs from endogenous mouse neurons. Please check whether ETZ, C3-DMA-DTZ, and DTZ differ in their ability to deliver DTZ into HEK293T cells in vitro and also include a direct comparison of these same prosubstrates in mice transduced with the AAV-delivered *hSyn1*-luciferase reporters.*

Response: We have performed the suggested HEK293T experiment (new Supplementary Fig. 4), showing that ETZ slows the bioluminescence decay kinetics and increases the overall photon production. Moreover, we compared the substrates using BREP AAV-transduced mice (new Supplementary Fig. 6). The ETZ group at the saturation substrate concentration (6.8 mM) still outperformed other groups for the long-lasting, brightest bioluminescence. At 2.5 mM (the saturation concentration for DTZ), the bioluminescence of the ETZ group was only marginally higher than the DTZ group (45% higher, but statistically insignificant). These results suggest that the enhanced *in vivo* performance of ETZ from DTZ was primarily caused by the increased substrate delivery enabled by the increased solubility. We have added the discussion to the manuscript.

Part of the rationale given for including a carboxylate in ETZ is to prevent BBB exclusion by Pgp. Please include discussion of ABCG2, which also operates at the BBB but has different substrate specificity.

Response: Thanks for pointing out that ABCG2 may also possibly pump out CTZ-like luciferins. We have revised our manuscript as follows: “the BBB efflux transporters, such as P-glycoprotein (P-gp) and ABCG2, were shown to pump out CTZ.^{33,34}”.

Decreasing lipophilicity was also proposed to increase brain delivery. The computed logP of DTZ is given as 4.3, but the logP values for ETZ and C3-DMA-DTZ are not described, and C3-DMA-DTZ performs significantly worse than DTZ. Please include these logP values and discuss. Are the results consistent with the hypothesis about decreasing lipophilicity? Or does it suggest the importance of other factors such as the rate of hydrolysis, binding to serum proteins, artifacts arising from the use of HEK293T cells, or something else? Please include other prosubstrate examples (such as the readily available compound 4) to gain better insight into the factors involved.

Response: Since ETZ and C3-DMA-DTZ are expected to be charged at physiological pH, we have now calculated and presented the logD values (the octanol/water distribution coefficient after considering molecules' ionization) of these compounds (4.5, 1.1, and 3.8 at pH 7 for DTZ, ETZ, and C3-DMA-DTZ, respectively). Also, in response to your early comment, we compared the substrates using BREP AAV-transduced mice (new Supplementary Fig. 6). The results suggest that the enhanced *in vivo* performance of ETZ from DTZ was primarily caused by the increased substrate delivery enabled by the increased solubility.

...AAV delivery into the brains of live mice was fruitfully applied to compare the performance of BRIC, BREP, and OCaMBI in vivo, but comparison of DTZ, ETZ, and C3-DMA-DTZ should also be performed in these mice....

Response: This comment is repeating an early comment, and please also see our response above. We have performed the suggested experiment.

The new compounds, particularly the prosubstrates ETZ and C3-DMA-DTZ, should have HRMS exact mass data

Response: The HR-MS data and spectra are now added to the Method section and the Supplementary Information.

Ref 39 describes AAV-PHP.eB et al. that can cross the BBB in several mouse strains, but notably not the BALB/c strain used in this manuscript.

Response: We have used ETZ in both BALB/c and C57BL/6J mice and gained high brightness from the mouse brain. Among the results presented in this manuscript, C57BL/6J mice, which are compatible with AAV-PHP.eB, were used for kainic acid-induced seizures (**Fig 4C**). In future studies, we plan to explore these new AAV vectors for peripheral gene delivery and noninvasive recording of neuronal activity in transgenic Cre mice with the C57BL/6 genetic background. Accordingly, we have expanded the discussion in the manuscript as follows: “This study tested ETZ in both BALB/cJ and C57BL/6J strains of mice Recently, AAV vectors with engineered capsids have been reported to cross the BBB of C57BL/6J mice and marmoset.^{55,56} We are currently exploring these new AAV vectors for peripheral gene delivery and transduction of specific neuronal populations in transgenic Cre mice with the C57BL/6 genetic background.”

“We named this new compound ETZ for esterase-dependent activation and enhanced in vivo performance”. Does the E stand for esterase or enhanced? Both are underlined.

Response: Both words (“esterase” and “enhanced”) made us name the new compound ETZ. Also, we think ETZ is an appropriate name since the new compound is developed after CTZ and DTZ.

“Photos” within the optical window should be “photons”.

Response: Corrected.

Reviewer 2

We appreciate these comments, which have helped up improve the paper significantly. We have performed additional experiments and revised our manuscript accordingly.

1) the authors description of the advantages/disadvantages of aequorin is inaccurate. Aequorin doesn't emit photons slowly---in fact, when activated by Ca⁺⁺, aequorin and obelin are very fast and can track action potentials. The major problem with aequorin for sensing neural activity over time is that aequorin discharges when activated by Ca⁺⁺ and it only slowly recharges in the presence of coelenterazine (also, it is blue emission which doesn't penetrate tissue well).

Response: Sorry for the confusion. We wanted to mention the low brightness, a.k.a. slow photon production rate (photons/enzyme molecule/s). We have now revised the sentence to avoid confusion: “Previous studies have used Ca²⁺-sensitive photoproteins, such as aequorin, obelin, and their mutants, for Ca²⁺ detection, **but they have drawbacks, including weak light emission and irreversible responses due to slow luciferin recharging.**²¹⁻²³” We don't discuss the blue emission here since Reference 22 has reported red-shifted aequorin variants.

2) Fig. 1D: No images of background luminescence controls are shown for the injection of non-BRIC expressing HEK cells, followed by the administration of DTZ and ETZ. This is important, especially since it appears that the researchers are injecting a higher concentration of ETZ than DTZ. For example, if the researchers inject 100uL of equal concentrations of ETZ and DTZ (2.5mM each instead of 6.8 and 2.5mM), do they still see an improvement in bioluminescence? Is the main advantage of the new substrate that a higher concentration can be injected due to the increased solubility?

Response: We previously investigated and confirmed the negligible background of DTZ in mice (see our previous publication *Biochemistry* 2019, 58 (12), 1689-1697). In our revision, we further examined ETZ in blank mice and mice intracranially injected with untransfected HEK 293T cells, reconfirming the negligible background of ETZ (new Supplementary Fig. 7). A previous *Science* publication presented the high background of DTZ, but our 2019 *Biochemistry* paper has clearly shown that the *Science* publication misinterpreted the results.

As for the second part of the comment, we have performed the additional comparison between 2.5 mM DTZ and 2.5 mM ETZ (new Supplementary Fig. 6). The results indeed suggest that the enhanced *in vivo* performance of ETZ from DTZ was primarily caused by the increased substrate delivery enabled by the increased solubility. We have revised the discussion of the manuscript.

3) Fig. 2: BRIC response is only 2X from 10e7 to 10e6 Ca⁺⁺, which is not a large dynamic range. If true, the authors need to acknowledge this less-than-optimal characteristic. Also, since this is an INTENSITY reporter (i.e., not a ratiometric reporter), it will therefore be subject to movement artifacts in a freely moving animal. Please acknowledge these limitations.

Response: Thanks for the suggestion. We have added the discussion: “.....**However, despite the enhancement, BRIC is still suboptimal since its response to physiological Ca²⁺ changes (e.g., from ~50 nM to ~1 μM) is limited to ~ 2-fold. We plan to further optimize its practical dynamic range within the physiological Ca²⁺ concentration range in the future**”; and “.....BRIC is an intensimetric indicator, and awake mice were head-fixed for macroscopic BLI. Unavoidable motions or blood flow changes in

animals may still induce small intensity changes, resulting in potential artifacts in recording small Ca^{2+} changes. Future developments may allow two-channel BLI of the mouse brain with high sensitivity, enabling ratiometric correction for baseline decays, blood flow changes, and motions of freely moving animals.”

4) Fig 2G: clarify that the measurement is of bioluminescence intensity (what wavelength range?) and not a ratiometric reading.

Response: All imaging studies were intensiometric ($\Delta\text{BL}/\Delta\text{BL}_0$) and no filter was used. The experimental conditions are mentioned in figure legends and/or the Method section. Btw, we didn't develop a red/teal ratiometric sensor, because the brain tissue, skull, and skin strongly attenuated short-wavelength light and we wouldn't have enough photons to perform macroscopic, red/teal ratiometric BLI of mice with intact skulls.

5) Supp. Fig. 3: add furimazine to this experiment for comparison. Even in the original paper (Yeh et al. 2017), I don't see a clear comparison of teLuc with furimazine.

Response: The luciferases and luciferins were reported as pairs (e.g., NanoLuc-furimazine, and teLuc-DTZ). Since the reviewer wanted to see cross-reactivity between teLuc-derived BREP and DTZ, we have performed additional experiments (new Supplementary Fig. 6). In the mouse model, the integrated brightness of BREP-ETZ was 122.7-fold higher than BREP-furimazine (substrates at their respective saturation concentrations).

6) Fig. 3: FFz is only tested with OCaMBI, but teLuc is also a NanoLuc variant. Therefore, BRIC needs to be tested with FFz too.

Response: Again, FFz was reported to be a new substrate paired with NanoLuc for *in vivo* imaging. Since the reviewer wanted to see its cross-reactivity with teLuc-derived luciferases, we have performed additional experiments (new Supplementary Fig. 6). We used AAV-BREP transduced mice to compare various substrates. In the mouse model, the integrated brightness of BREP-FFz was 67.7-fold higher than BREP-furimazine (substrates at their respective saturation concentrations).

7) Fig 4B: generally the foot shock data are very nice. Can the authors compare the kinetics of BRIC with those of GCaMP, which is the main competing technology? For example, the authors should inject GCaMP into the amygdala & perform fiber photometry w/ 500ms exposure with an EMCCD camera, and compare the width of the response to that of their BRIC sensor.

Response: We are sorry for the miscommunication regarding the focus of our work. Our goal here is to develop enabling technology for imaging the activity of **neuronal ensembles (populations)** in awake mice **minimally invasively**. Our new approach is unlikely to increase spatiotemporal resolution from fluorescence or electric recording. Fluorescence remains the choice of imaging modality for fiber photometry or single-cell-resolution imaging through cranial windows. On the other hand, our new method will offer technical simplicity, noninvasiveness, and deep-brain imaging capability. The new approach will complement, but cannot replace, GCaMP-based fluorescence imaging. We have added the discussion to the revised manuscript to reduce further confusion.

8) Fig 4B: what do the raw traces look like over the entire recording relative to the start of the substrate injection? From Fig 3C it seems like there is a fast decay in signal from the time of the injection, so it is important to determine how that affects the size of the responses to the foot shocks.

Response: Here is a raw trace (residual background of ~480 unsubtracted) for a BRIC-expressing mouse in response to 13 consecutive trials of footshock stimulations. In the Fig. 4 legend, we mentioned that the baselines for all intensity traces were corrected using a monoexponential decay model. The Method section includes more details on data analysis for individual experiments. Please note that we present the data as intensity ratios ($\Delta BL / \Delta BL_0$), so as long as the traces are properly corrected for residual background and baseline decay, the overall intensity decay shouldn't affect the size of the responses presented as intensity ratios (meanwhile, the signal-to-noise ratio would become worse as signals decay).

9) Fig 4C: why is the calcium activity so variable across the mice for seizure? Is this an issue with the assay for detecting calcium, or is it a problem with the substrate not working consistently when injected?

Response: The failure to inject the substrate would be manifested as very low bioluminescence signals, which were not observed when we performed the experiment in Fig. 4C. We reasoned that the variation in the extent of BRIC signal changes was caused by the following two reasons: (1) intrinsic biological variations since the level of KA-induced seizures and calcium responses could be different in individual animals, and (2) technical limitations since KA-induced seizures are known to be intermittent and some mice were probably not imaged right during the occurrence of seizures. We used the tremor behavior of mice as an indicator to initiate ETZ injection and BLI, but seizures may stop intermittently during the 12 min imaging period. We have added the discussion to the revised manuscript. Btw, we could manually exclude some BRIC mice from the analysis, but we want to minimize human intervention, which may sometimes be biased. Nevertheless, there was a clear statistical difference between BRIC- and BREP-expressing mice ($P = 0.0057$ and 0.0295 for peak response and AUC, respectively).

10) Fig 4B/C: there appears to be a small (but significant) increase in activity in the calcium-independent BREP when the animals receive a foot shock in 4B, and when the animals are undergoing seizure in 4C. What is causing that? Motion? Increased blood flow due to neural activity? Does this limit the usefulness of the sensor system if there are potentially small calcium-independent changes in bioluminescence with strong neural activity/motion? Address these limitations in the Discussion.

Response: Thanks for the suggestion. The bioluminescence images were acquired macroscopically. Thus, motions or blood flow changes in awake animals might cause small intensity changes. We indeed consider this as a potential limitation and have added the discussion to the manuscript.

11) Add Reference(s) to the Introduction to support the important points that BLI is superior to fluorescence in terms of photobleaching, phototoxicity, long-term measurements (e.g., circadian rhythms), and as optogenetic partners.

Response: Thanks for the suggestion. We have added new references for these claims.

12) The manuscript would benefit from some minor editing of English to enhance clarity.

Response: We have performed further proofreading and made corrections.

Reviewer 3

We thank Review 3 for supportive comments, as the tools were evaluated to be “of great interest to many in the community” and the manuscript was considered “an important step towards this future by markedly improving brain delivery of luciferins and by improving the dynamic range and signal strength of BL calcium sensors”. A few minor concerns were raised, which have been addressed as discussed below:

.....The authors should be cautious about overstating the advantages of the described BRIC sensor. While certainly a notable improvement over previous generations of BL calcium sensor, BRIC is still suboptimal in several ways. While the *in vitro* characterization of the sensor shows a ~6-fold dynamic range, the practical dynamic range is quite a bit smaller, considering that the K_d for calcium binding is ~133nM and the Hill coefficient appears to be fairly low as well, and so the signal change within the physiological range (~50nM to ~1 μ M) is in fact considerably smaller than the full dynamic range. This is the likely reason that HeLa cells generated only ~2-fold changes in BL intensity with calcium oscillations.

Response: Thank you for the suggestion. We have added the following discussion: “However, despite the enhancement, BRIC is still suboptimal since its response to physiological Ca^{2+} changes (e.g., from ~50 nM to ~1 μ M) is limited to ~ 2-fold. We plan to further optimize its practical dynamic range within the physiological Ca^{2+} concentration range in the future.”

When characterizing the BRIC sensor in cultured neurons, a more gentle stimulation than depolarization with high KCl would be a welcome addition to this manuscript. Ideally, titration of the sensor in living cells (with CaEGTA buffers and ionomycin, for example) would provide very useful information about the differences in sensor behavior in vitro vs. in cellulo. However, I accept the evidence shown in intact mice subjected to foot shock as a legitimate demonstration of the sensor's basic functionality.

Response: As you suggested, we attempted to titrate BRIC-expressing neurons with Ca^{2+} and ionomycin but soon realized that we couldn't figure out the exact concentrations of Ca^{2+} brought into cells by ionomycin. Nevertheless, we were able to show the responsiveness of BRIC in primary neurons to ionomycin-induced Ca^{2+} influx. These results are presented in the new Supplementary Fig. 11.

In general, the authors have provided sufficient evidence for most claims made in this manuscript. However, it should be noted that while it appears superficially that KA-induced seizures do indeed lead sometimes to detectable changes in BL intensity from the BRIC sensor, the variance in these experiments was extremely high and the authors are clearly struggling to find a condition that reaches statistical significance. The data presented in Fig. 4C is unfortunately not entirely convincing, since some BRIC animals showed essentially no response. This might be solved by adding more animals, but my suspicion is that this seizure model simply doesn't produce strong enough calcium signals over a large enough area to work with the sensor as it currently stands (likely because its K_d and Hill coefficient are both too low, or because the population of neurons generating high calcium signals is too small). I suggest removing this from the manuscript, as it is really not necessary to demonstrate the utility of the sensor in its current form. Alternatively, the authors could include further discussion about this experiment and why the results are less robust than those from other experiments in this study.

Response: We reasoned that the variation in the extent of BRIC signal changes was caused by the following two reasons: (1) intrinsic biological variations since the level of KA-induced seizures and calcium responses could be different in individual animals, and (2) technical limitations since KA-induced seizures are known to be intermittent and some mice were probably not imaged right during the occurrence of seizures. We used the tremor behavior of mice as an indicator to initiate ETZ injection and BLI, but seizures could stop intermittently during the 12 min imaging period. For the latter reason, we could manually exclude some data from the analysis, but we want to minimize human intervention, which may sometimes be biased. Nevertheless, there was a clear statistical difference between BRIC- and BREP-expressing mice ($P = 0.0057$ and 0.0295 for peak response and AUC, respectively).

We have added a discussion on possible reasons for the experimental variation. We also discussed the limitations of the BRIC sensor (e.g., responses to physiological Ca^{2+} changes, a requirement of concerted changes in a neuronal population). We prefer to keep the Fig. 4C result in the manuscript since this experiment used C57BL/6J, supporting the potential of using AAV vectors with engineered capsids (e.g., AAV-PHP.eB) to noninvasively image the brain activity of transgenic Cre mice.

The authors should consider clarifying some of the limitations of the current system in terms of spatial and temporal resolution. A brief list of potential biological questions that could be addressed using the BRIC/ETZ system would enhance the manuscript quite a bit. It appears that exposure times for all imaging in this study are between 500ms and 2s, a fairly major limitation when many calcium signals occur on the order of tens of

milliseconds. It's important to acknowledge these limitations in the discussion section at some point more clearly, if only to keep expectations realistic among potential end users of this technology. What varieties of brain activity occur on achievable time and spatial scales?

Response: Thank you for the suggestion. We have added the following discussion: “Our current approach macroscopically records concerted changes in a neuronal population that expresses the BRIC indicator. The spatial resolution is limited by scattering when photons travel through brain tissue, skull, and skin. Also, photons reaching the detector are still scarce, and when using the experiment setups presented here, the temporal resolution is in the second and subsecond range. The overall duration of the rising and extinction of intracellular Ca^{2+} signals after the action potential is typically in this time range, but the kinetics of Ca^{2+} concentration changes is much faster than our achieved temporal resolution.⁵⁴ Thus, interested users should be aware that the recorded signals are undersampled over time. Considering these reasons and the fact that fluorescent Ca^{2+} indicators, such as GCaMPs, have been extensively optimized,⁵⁴ fluorescence remains the choice of imaging modality for fiber photometry or single-cell-resolution recording through cranial windows. On the other hand, our new method offers technical simplicity, reduced invasiveness, and deep-brain imaging capability. The new approach complements, but cannot replace, fluorescence or electrical recording.”

..... An additional supplementary figure including the chemical structures of other luciferins used or mentioned in the manuscript would be very helpful (e.g. furimazine and FFz along side DTZ, ETZ, C2-DMA-DTZ for comparison). There are minor typographical errors throughout that need to be corrected, but no major issues with readability or content of the manuscript...

Response: We have added a new Supplementary Fig. 1 to include the chemical structures. Also, we have further proofread the manuscript and made corrections.

Reviewers' Comments:

Reviewer #1:

Remarks to the Author:

The authors have revised their manuscript and were largely responsive to the reviewer comments. The work should be of broad interest and is worthy of publication. However, in light of the data that the increased solubility and dosage of ETZ are the main factors for the enhanced performance, the statements in the abstract and discussion should be modified to reflect the data. In the abstract, it is stated that "we developed a luciferase prosubstrate activatable in vivo by nonspecific esterase to enhance the brain delivery of the luciferin". The word "brain" should be removed as the "brain delivery" was not enhanced in an apples-to-apples comparison to DTZ. Similarly, in the discussion it is stated that "we first enhanced the brain delivery", when the data shows that it is the solubility and dosage that was enhanced, not the ability to access the brain. Finally, the discussion could use a little refinement. Perhaps owing to the changes made in the manuscript in response to the reviewer comments, it seems a bit disjointed.

Reviewer #2:

Remarks to the Author:

The authors have adequately responded to our concerns about the original submission.

Reviewer #3:

Remarks to the Author:

The authors have adequately addressed my minor concerns as well as many additional concerns by the other reviewers. This manuscript should be accepted.

Reviewer 1

The authors have revised their manuscript and were largely responsive to the reviewer comments. The work should be of broad interest and is worthy of publication. However, in light of the data that the increased solubility and dosage of ETZ are the main factors for the enhanced performance, the statements in the abstract and discussion should be modified to reflect the data. In the abstract, it is stated that "we developed a luciferase prosubstrate activatable in vivo by nonspecific esterase to enhance the brain delivery of the luciferin". The word "brain" should be removed as the "brain delivery" was not enhanced in an apples-to-apples comparison to DTZ. Similarly, in the discussion it is stated that "we first enhanced the brain delivery", when the data shows that it is the solubility and dosage that was enhanced, not the ability to access the brain. Finally, the discussion could use a little refinement. Perhaps owing to the changes made in the manuscript in response to the reviewer comments, it seems a bit disjointed.

Response: We have now rephrased these sentences. We have also refined the discussion section to improve the flow. We think the current manuscript has addressed these concerns.

Reviewer 2 & Reviewer 3 have no additional concern.